# Offline Inverse Constrained Reinforcement Learning for Safe-Critical Decision Making in Healthcare

## Abstract

Reinforcement Learning (RL) applied in healthcare can lead to unsafe medical decisions and treatment, such as excessive dosages or abrupt changes, often due to agents overlooking common-sense constraints. Consequently, Constrained Reinforcement Learning (CRL) is a natural choice for safe decisions. However, specifying the exact cost function is inherently difficult in healthcare. Recent Inverse Constrained Reinforcement Learning (ICRL) is a promising approach that infers constraints from expert demonstrations. ICRL algorithms model Markovian decisions in an interactive environment. These settings do not align with the practical requirement of a decision-making system in healthcare, where decisions rely on historical treatment recorded in an offline dataset. To tackle these issues, we propose the Constraint Transformer (CT). Specifically, 1) we utilize a causal attention mechanism to incorporate historical decisions and observations into the constraint modeling, while employing a Non-Markovian layer for weighted constraints to capture critical states. 2) A generative world model is used to perform exploratory data augmentation, enabling offline RL methods to simulate unsafe decision sequences. In multiple medical scenarios, empirical results demonstrate that CT can capture unsafe states and achieve strategies that approximate lower mortality rates, reducing the occurrence probability of unsafe behaviors.

## 1 Introduction

In recent years, the doctor-to-patient ratio imbalance has drawn attention, with the U.S. having only 223.1 physicians per 100,000 people (Petterson et al., 2018). AI-assisted therapy emerges as a promising solution, offering timely diagnosis, personalized care, and reducing dependence on experienced physicians. Therefore, the development of an effective AI healthcare assistant is crucial.

Reinforcement learning (RL) offers a promising approach to develop AI assistants by addressing sequential decision-making tasks. However, this method can still lead to unsafe behaviors, such as administering excessive drug dosages, inappropriate adjustments of medical parameters, or abrupt changes in medication dosages. These actions, including **"too high"** or **"sudden change"**, may significantly endanger patients, potentially resulting in acute hypotension, arrhythmias, and organ damage, with fatal consequences (Jia et al., 2020; Shi et al., 2020). For example, in sepsis treatment, vasopressor (vaso) doses above $1\mu g/(kg \cdot min)$ are linked to a 90% mortality rate (Martin et al., 2015), and sudden changes in vaso can cause dangerous blood pressure fluctuations (Fadale et al., 2014). Our experiments show that Huang et al.

Table 1: Proportion of unsafe vaso doses recommended by physician and DDPG policy. Typical vaso dosages range from 0.1 to $0.2\mu g/(kg \cdot min)$, with doses above 0.5 considered high (Bassi et al., 2013). A critical threshold of 0.75 is associated with increased mortality (Auchet et al., 2017).

| Actions ($\mu g/(kg \cdot min)$) | Physician policy | DDPG policy |
|---|---|---|
| vaso >0.75 | 2.27% | 7.44% ↑ |
| vaso >0.9 | 1.71% | 7.40% ↑ |
| $\Delta$ vaso >0.75 | 2.45% | 21.00% ↑ |
| $\Delta$ vaso >0.9 | 1.88% | 20.62% ↑ |

$\Delta$ vaso: The change in vaso doses between two-time points.
↑: There is a high proportion of unsafe actions under this policy.

(2022) use of the DDPG algorithm in sepsis, which exhibits **"too high"** [1] and **"sudden change"** [2] in vaso recommendations, as seen in Table 1. Moreover, if the dosage is clipped using simple thresholding, it will not account for the individualized tolerance of each patient.

---

[1] "too high" refers to a lethal drug dose for a particular patient; however, this is not a single exact value, as it can vary depending on the patient's individual condition. Analysis of the condition, see the Appendix B

[2] "sudden change" indicates that the change in dosage between two-time points exceeds the threshold.

This paper aims to achieve safe healthcare policy learning to mitigate unsafe behaviors. The most common method for learning safe policies is Constrained Reinforcement Learning (CRL) (Liu et al., 2021; 2022), with the key to its success lying in the constraints representation. However, in healthcare, we can only design the cost function based on prior knowledge, which limits its application due to a lack of personalization, universality, and reliance on prior knowledge. For more details about issues, please refer to Appendix C. Inverse Constrained Reinforcement Learning (ICRL) (Malik et al., 2021) emerges as a promising approach, as it can infer the constraints adhered to by experts from their demonstrations. However, existing ICRL methods face the following challenges in healthcare:

**1) The Markov decision [3] is not compatible with medical decisions.** ICRL algorithms model Markov decisions, where the next state depends only on the current state and not on the history (Kijima, 2013; Zhang et al., 2023). However, in healthcare, the historical states of patients are crucial for medical decision-making (Plaisant et al., 1996). Therefore, ICRL algorithms based on Markov assumption can not capture patient history, and ignore individual patient differences, thereby limiting effectiveness. **2) Interactive environment is not available for healthcare or medical decisions.** ICRL algorithms (Malik et al., 2021; Gaurav et al., 2022) follow an online learning paradigm, allowing agents to explore and learn from interactive environments. However, exploration in healthcare often entails unsafe behaviors that could breach constraints and result in substantial losses. Therefore, it is necessary to infer constraints using only offline datasets.

In this paper, we introduce offline Constraint Transformer (CT), a novel ICRL framework that incorporates patients' historical information into constraint modeling and learns from offline data to infer constraints in healthcare. Specifically,

1) Inspired by the recent success of sequence modeling (Zheng et al., 2022; Chen et al., 2021; Kim et al., 2023), we incorporate historical decisions and observations into constraint modeling using a causal attention mechanism. To capture key events in trajectories, we introduce a Non-Markovian transformer to generate constraints and cost weights, and then define constraints using weighted sums. CT takes trajectories as input, allowing for the observation of patients' historical information and evaluation of key states.

2) To learn from an offline dataset, we introduce a model-based offline RL method that simultaneously learns a policy model and a generative world model via auto-regressive imitation of the actions and observations in medical decisions. The policy model employs a stochastic policy with entropy regularization to prevent it from overfitting and improve its robustness. Utilizing expert datasets, the generative world model uses an auto-regressive exploration generation paradigm to effectively discover a set of violating trajectories. Then, CT can infer constraints in healthcare through these unsafe trajectories and expert trajectories.

In the medical scenarios of sepsis and mechanical ventilation, we conduct experimental evaluations of offline CT. Experimental evaluations demonstrate that offline CT can capture patients' unsafe states and assign higher penalties, thereby providing more interpretable constraints compared to previous works (Huang et al., 2022; Raghu et al., 2017a; Peng et al., 2018). Compared to unconstrained, custom constraints and LLMs constraints (designed by Large Language Models (LLMs)), CT achieves strategies that closely approximate lower mortality rates with a higher probability (improving by $8.85\%$ compared to DDPG). To investigate the avoidance of unsafe behaviors with offline CT, we evaluate the probabilities of "too high" and "sudden changes" occurring in the sepsis. The experimental results show that CRL with CT can reduce the probability of unsafe behaviors to zero.

## 2 RELATED WORKS

**Reinforcement Learning in Healthcare.** RL has made great progress in the realm of healthcare, such as sepsis treatment (Huang et al., 2022; Raghu et al., 2017a; Peng et al., 2018; Do et al., 2020), mechanical ventilation (Kondrup et al., 2023; Gong et al., 2023; Yu et al., 2020), sedation (Eghbali et al., 2021) and anesthesia (Calvi et al., 2022; Schamberg et al., 2022). However, the works mentioned above have not addressed potential safety issues such as sudden changes or too high medications. Therefore, the development of policies that are both safe and applicable across various healthcare domains is crucial.

---

[3]Markov decision generally refers to first-order Markov.

**Inverse Constrained Reinforcement Learning.** Previous works inferred constraint functions by determining the feasibility of actions under current states. In discrete state-action space, Chou et al. (2020) and Park et al. (2020) learned constraint sets to differentiate constrained state-action pairs. Scobee & Sastry (2019) proposed inferring constraint sets based on the principle of maximum entropy, while some studies (McPherson et al., 2021; Baert et al., 2023) extended this approach to stochastic environments using maximum causal entropy (Ziebart et al., 2010). However, discrete approaches often face limitations when scaling to high-dimensional problems. As the state-action space increases, the computational cost rises significantly. This makes inference in large, discrete spaces challenging, requiring additional optimization techniques or assumptions. In continuous domains, Malik et al. (2021), Gaurav et al. (2022), and Qiao et al. (2024) used neural networks to approximate constraints. Some works (Liu et al., 2022; Chou et al., 2020) applied Bayesian Monte Carlo and variational inference to infer the posterior distribution of constraints in high-dimensional state space. Xu & Liu (2023) modeled uncertainty perception constraints for arbitrary and epistemic uncertainties. However, these methods can only be applied online and lack historical dependency.

**Transformers for Reinforcement Learning.** Transformer has produced exciting progress on RL sequential decision problems (Zheng et al., 2022; Chen et al., 2021; Janner et al., 2021; Liu et al., 2023). These works no longer explicitly learn Q-functions or policy gradients, but focus on action sequence prediction models driven by target rewards. Chen et al. (2021) and Janner et al. (2021) perform auto-regressive trajectories modeling to achieve offline policy learning. Furthermore, Zheng et al. (2022) unify offline pretraining and online fine-tuning within the Transformer framework. Liu et al. (2023) and Kim et al. (2023) integrate the transformer architecture into constraint learning and preference learning. With its sequence modeling capability and independence from the Markov assumption, the transformer architecture can capture temporal dependencies in medical decision-making. Thus, it is well-suited for trajectory learning and personalized learning in medical settings.

## 3 PROBLEM FORMULATION

**Constrained Reinforcement Learning (CRL).** We model the medical environment with a Constrained Partially Observable Markov Decision Process (Constrained POMDP) $\mathcal{N}^c$, which can be defined by a tuple $(\mathcal{S}, \mathcal{A}, \mathcal{O}, \mathcal{P}, \mathcal{R}, \mathcal{C}, \Omega, \gamma, \kappa, \rho_0, T)$ where: 1) $s \in \mathcal{S}$ represents the unobservable true state indicators of the patient at each time step. 2) $a \in \mathcal{A}$ corresponds to the administered drug doses or instrument parameters of interest. 3) $o \in \mathcal{O}$ represents the observable patient indicators (e.g., vital signs, lab results) at each time step. These observations partially reflect the true state $s_t$. 4) $\mathcal{P}(s_{t+1} \mid s_t, a_t)$ defines the transition probabilities. 5) The reward function $\mathcal{R}(s_t, a_t)$ is used to describe the quality of the patient's condition and provided by experts based on prior work (Huang et al., 2022; Kondrup et al., 2023). 6) The constraint function $\mathcal{C}(s_t, a_t)$ describes the risk or cost associated with taking a particular action given the current historical information. 7) The observation probability function $\Omega(o \mid s, a)$ defines the probability of observing $o \in \mathcal{O}$ given the true state $s \in \mathcal{S}$ and the action $a \in \mathcal{A}$. 8) $\gamma$ denotes the discount factor. 9) $\kappa \in \mathbb{R}_+$ denotes the bound of cumulative costs. 10) $\rho_0$ defines the initial state distribution. 11) $T$ is the length of the trajectory $\tau$. At each time step $t$, an agent acts $a_t$ at a patient's state $s_t$. This process generates the reward $r_t \sim \mathcal{R}(s_t, a_t)$, the cost $c_t \sim \mathcal{C}(s_t, a_t)$ and the next state $s_{t+1} \sim \mathcal{P}(s_{t+1} \mid s_t, a_t)$. The goal of the CRL policy $\pi$ is to maximize expected discounted rewards while limiting the cost in a threshold $\kappa$:

$$\arg\max_\pi \mathbb{E}_{\pi, \rho_0}\left[\sum_{t=1}^T \gamma^t r_t\right], \quad \text{s.t.} \quad \mathbb{E}_{\pi, \rho_0}\left[\sum_{t=1}^T \gamma^t c_t\right] \leq \kappa. \tag{1}$$

CRL commonly assumes that constraint signals are directly observable. However, in healthcare, such signals are often difficult to obtain due to the variability in individual patient characteristics and the need for multi-objective evaluation. Therefore, our objective is to infer reasonable constraints for CRL to achieve safe policy learning in healthcare.

**Inverse Constrained Reinforcement Learning (ICRL).** ICRL (Malik et al., 2021) based on Markov Decision Process (MDP) $\mathcal{M}$ aims to recover the cost function $\mathcal{C}^*$ by leveraging a set of trajectories $\mathcal{D} = \{\tau^{(i)}\}_{i=1}^N$ sampled from an expert policy $\pi_e$, where $N$ denotes the number of the trajectories. ICRL is commonly based on the Maximum Entropy framework (Scobee & Sastry, 2019), and the likelihood function is articulated as:

$$p(\mathcal{D} \mid \mathcal{C}) = \frac{1}{Z^N} \prod_{i=1}^N \exp\left[\mathcal{R}(\tau^{(i)})\right] \mathbb{I}(\tau^{(i)}) \tag{2}$$

Here, 1) $\tau = \{s_0, a_0, s_1, ...\}$ is the trajectory from $\mathcal{D}$. 2) $Z = \int \exp(\beta r(\tau)) \mathbb{I}(\tau) d\tau$ is the normalizing term, where $\beta \in [0, \infty)$ is a parameter that measures the proximity of the agent to the

optimal distribution. As $\beta \to \infty$, the agent approaches optimal behavior, whereas as $\beta \to 0$, the agent's actions become increasingly random. 3) The indicator $\mathbb{I}(\tau^{(i)})$ signifies the extent to which the trajectory $\tau^{(i)}$ satisfies the constraints. It can be approximated using a neural network $\zeta_\theta(\tau^{(i)})$ parameterized with $\theta$, defined as $\zeta_\theta(\tau^{(i)}) = \prod_{t=0}^{T} \zeta_\theta(s_t^{(i)}, a_t^{(i)})$. Consequently, the cost function can be formulated as $C_\theta = 1 - \zeta_\theta$. Substituting the neural network for the indicator, we can update $\theta$ through the gradient of the log-likelihood function:

$$\nabla_\theta \mathcal{L}(\theta) = \mathbb{E}_{\tau \sim \pi_e} \left[ \nabla_\theta \log[\zeta_\theta(\tau)] \right] - \mathbb{E}_{\hat{\tau}} \sim \pi_{\mathcal{M}^{\hat{\zeta}_\theta}} \left[ \nabla_\theta \log[\zeta_\theta(\hat{\tau})] \right] \tag{3}$$

where $\tau$ is sampled from the expert policy $\pi_e$, while $\hat{\tau}$ is sampled from the executing policy $\pi_{\mathcal{M}^{\hat{\zeta}_\theta}}$. $\mathcal{M}^{\hat{\zeta}_\theta}$ denotes the MDP derived by augmenting the original MDP with the network $\hat{\zeta}_\theta$. The policy $\pi_{\mathcal{M}^{\hat{\zeta}_\theta}}$ is used to execute this augmented MDP.

**Safe-Critical Decision Making with Constraint Inference in Healthcare.** Our general goal for our policy is to align the expert policy, which refers to the strategy under which the patient's mortality rate is minimized (achieving a zero mortality rate is often difficult since there are patients who can not recover, regardless of all potential future treatment sequences (Fatemi et al., 2021)). Decision-making with constraints can formulate safer strategies by discovering and avoiding unsafe states, thereby aligning the expert policy.

However, most offline RL algorithms rely on online evaluation, where the agent is evaluated in an interactive environment, whereas in medical scenarios, only offline policy evaluation (Luo et al., 2024a) can be utilized. Besides, some works (Jia et al., 2020; Huang et al., 2022; Raghu et al., 2017b; Komorowski et al., 2018) analyze by comparing the differences (DIFF) between the drug dosage recommended by the estimated policy $\pi$ and the dosage administered by clinical physicians $\hat{\pi}$, and the relationship of DIFF with mortality rates, through graphical analysis. In the graph depicting the relationship between the DIFF and mortality rate, at the point when DIFF is zero, the lower the mortality rate of patients, the better the performance of the policy (Raghu et al., 2017b).

To provide a more accurate quantitative evaluation, we introduce the concept of the aligning rate with the expert policy, defined as $\omega$:

$$\omega = \frac{\text{Number of survivors among the top } N \text{ patients}}{N} \tag{4}$$

1) We assume that the policy used to treat surviving patients in the medical dataset is the expert policy. We randomly select $2N$ patients from the dataset, where $N$ patients survived under the expert policy, and $N$ patients died under the non-expert policy. 2) For each patient in the real dataset, we have access to their state and the drug dosage administered by the doctor ($a$). Using an estimated policy, we compute an alternative drug dosage ($b$) for the same patient state. 3) For each patient, we calculate the difference between the dosages, defined as DIFF $= b - a$. This gives us $2N$ DIFF values across all patients. 4) We then sort the $2N$ DIFF values in ascending order. Next, we examine the survival status of the top $N$ patients based on the sorted DIFF values. 5) The top $N$ patients represent those for whom the difference between our policy and the expert policy is smallest. If the survival rate of these top $N$ patients (denoted as $\omega$) is higher, it suggests that our policy has a higher aligning rate with the expert. 6) Additionally, we need to account for the magnitude of the DIFF values. For patients who survived, a smaller DIFF value is more desirable, as it indicates a closer alignment between our policy and the doctor's policy.

## 4 METHOD

To infer constraints and achieve safe decision-making in healthcare, we introduce the Offline Constraint Transformer (shown in Figure 1), a novel ICRL framework.

In practice, ICRL can be conceptualized as a bi-level optimization task (Liu et al., 2022). We can 1) update this policy based on Equation 1, and 2) employ Equation 3 for constraint learning. Intuitively, the objective of Equation 3 is to distinguish between trajectories generated by expert policies and imitation policies that may violate the constraints. Specifically, **task 1** involves updating the policy using advanced CRL methods. Significant progress has been made in some works such as BCQ-Lagrangian (BCQ-Lag) (Fujimoto et al., 2019), COpiDICE (Lee et al., 2022), VOCE (Guan et al., 2024), and CDT (Liu et al., 2023). **Task 2** focuses on learning the constraint function, as shown in Figure 1. Our research primarily improves the latter process, addressing two key challenges

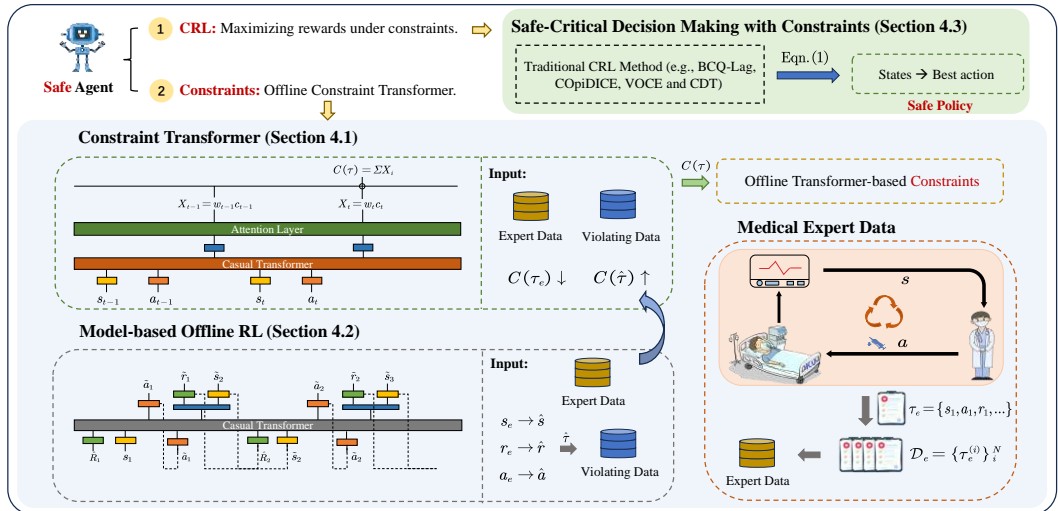

Figure 1: The overview of the safe healthcare policy learning with offline CT.

that ICRL faces in healthcare: **challenge 1** pertains to the limitations of the Markov property, and **challenge 2** involves the issue of inferring constraints only from offline datasets. To address these challenges, we propose the offline CT as our solution.

**Offline Constraint Transformer.** To address the first challenge, we delve into the inherent issues of applying the Markov property to healthcare and draw inspiration from sequence modeling tasks, redefining the representation of the constraints. To realize the offline training, we consider the essence of ICRL updates, proposing a model-based RL to generate unsafe behaviors used to train CT. We outline three parts: establishing the constraint representation model (Section 4.1), creating an offline RL for violating data (Section 4.2), and learning safe policies (Section 4.3).

## 4.1 CONSTRAINT TRANSFORMER

ICRL methods relying on the Markov property overlook patients' historical information, focusing only on the current state. However, both current and historical states, along with vital sign changes are crucial for a human doctor's decision-making process (Plaisant et al., 1996). To emulate the observational approach of humans, we draw inspiration from the existing historical sequence model (such as Long Short-Term Memory (LSTM) (Graves & Graves, 2012) and Transformer (Vaswani, 2017)) to incorporate historical

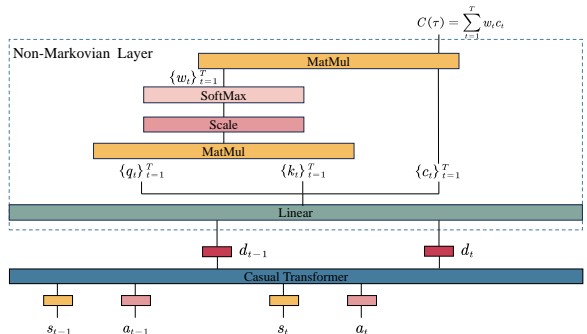

Figure 2: The structure of the CT.

information into constraints for a more comprehensive observation and judgment. Compared to LSTM, the Transformer effectively captures long-range dependencies and complex time series patterns due to its self-attention mechanism (Chen et al., 2023), without the need for sequential processing, which improves computational efficiency. Additionally, the dynamic attention weights in Transformers enhance model interpretability by highlighting the relative importance of different input elements. Therefore, we propose a constraint modeling approach based on a causal attention mechanism, as shown in Figure 2. The structure comprises a causal Transformer for sequential modeling and a Non-Markovian layer for weighted constraints learning.

**Sequential Modeling for Constraints Inference.** For a trajectory segment of length $T$, $2T$ input embeddings are generated, with each position containing state $s$ and action $a$ embeddings, which are learned by a linear layer and a normalization layer. And the state and action at the same timestep share the same positional embedding, which is also learned. Then the input embeddings are fed into the causal Transformer, which produces output embeddings $\{d_t\}_{t=0}^{T}$ determined by preceding input embeddings from $\{s_0, a_0, ..., s_T, a_T\}$. Here, $d_t$ depends only on the previous $t$ states and actions.

**Modeling Non-Markovian for Weighted Constraints Learning.** Although $d_t$ represents the cost function $c_t$ derived from observations over long trajectories, it does not pinpoint which previous key actions or states led to its increase. In healthcare, identifying key actions or states is vital for analyzing risky behaviors and status, and enhancing model interpretability. To address this, we draw inspiration from the design of the preference attention layer in (Kim et al., 2023) and introduce an additional attention layer. This layer is employed to define the cost weight for Non-Markovian. It takes the output embeddings $\{d_t\}_{t=0}^T$ from the casual transformer as input and generates the corresponding cost and the cost weights. The output of the attention layer (i.e., the cost function) is computed by weighting the values through the normalized dot product between the query and other keys:

$$C(\tau) = \frac{1}{T} \sum_{i=0}^{T} \sum_{t=0}^{T} \mathrm{softmax}\left(\{\langle q_i, k_{t'} \rangle\}_{t'=0}^T\right)_t \cdot c_t = \sum_{t=0}^{T} w_t \cdot c_t \tag{5}$$

Here, 1) the key $k_t \in \mathbb{R}^m$, query $q_t \in \mathbb{R}^m$, and value $c_t \in \mathbb{R}$ are derived from the $t$-th input $d_t$ through linear transformations, where $\mathbb{R}$ represents the set of real numbers and $m$ denotes the embedding dimension. Since $d_t$ depends only on the previous state-action pairs $\{(s_i, a_i)\}_{i=0}^t$ and serves as the input embedding for the Non-Markovian Layer, $c_t$ is also associated solely with the preceding $t$ time steps. 2) $w_t = \frac{1}{T} \sum_{i=0}^{T} \mathrm{softmax}\left(\{\langle q_i, k_{t'} \rangle\}_{t'=0}^T\right)_t$ depends on the full sequence $\{(s_t, a_t)\}_{t=0}^T$ to model the cost importance weight. Introducing the newly defined cost function, we redefine Equation 3 for CT as:

$$\nabla_\phi \mathcal{L}(\phi) = \mathbb{E}_{\tau_v \sim \mathcal{D}_v}\left[\nabla_\phi \log[C_\phi(\tau_v)]\right] - \mathbb{E}_{\tau_e \sim \mathcal{D}_e}\left[\nabla_\phi \log[C_\phi(\tau_e)]\right] \tag{6}$$

where $\phi$ is the parameter of CT. $\mathcal{D}_e$ and $\mathcal{D}_v$ represent the expert data and the violating data, respectively, while $\tau_e$ and $\tau_v$ are the trajectories from these datasets. This formulation implies that the constraint should be minimized on the expert policy and maximized on the violating policy. We construct an expert dataset and a violating dataset to evaluate Equation 6 offline. The expert data can be acquired from existing medical datasets or hospitals. Regarding the violating dataset, we introduce a generative model to establish it, as detailed in Section 4.2.

### 4.2 MODEL-BASED OFFLINE RL

To train CT offline, we introduce a model-based offline RL method (shown in Figure 3) to generate violating data that refers to unsafe behavioral data and can be represented as $\tau_v = \{s_0, a_0, r_0, s_1, ...\} \in \mathcal{D}_v$. The model simultaneously learns a policy

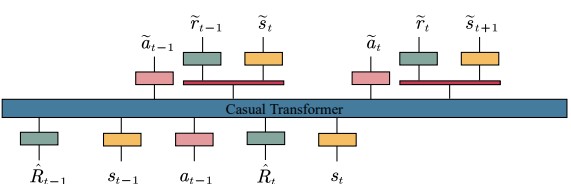

Figure 3: The structure of the model-based offline RL.

model and a generative world model via auto-regressive imitation of the actions and observations in healthcare. The model processes a trajectory, $\tau_e \in \mathcal{D}_e$, as a sequence of tokens encompassing the return-to-go, states, and actions, defined as $\{\hat{R}_0, s_0, a_0, ..., \hat{R}_T, s_T, a_T\}$. Notably, the return-to-go $\hat{R}_t$ at timestep $t$ is the sum of future rewards, calculated as $\hat{R}_t = \sum_{t'=t}^T r_{t'}$. At each timestep $t$, it employs the tokens from the preceding $K$ timesteps as its input, where $K$ represents the context length. Thus, the input tokens for it at timestep $t$ are denoted as $h_t = \{\hat{R}_{-K:t}, s_{-K:t}, a_{-K:t-1}\}$, where $\hat{R}_{-K:t} = \{\hat{R}_K, ..., \hat{R}_t\}$, $s_{-K:t} = \{s_K, ..., s_t\}$ and $a_{-K:t-1} = \{a_K, ..., a_{t-1}\}$.

**Policy Model.** The input tokens are encoded through a linear layer for each modality. Subsequently, the encoded tokens pass through a casual transformer to predict future action tokens. To explore a diverse set of actions and improve performance, we employ a stochastic Gaussian policy (Liu et al., 2023). Furthermore, we incorporate a Shannon entropy regularizer $\mathcal{H}[\pi_\vartheta(\cdot \mid h)]$ to prevent policy overfitting and enhance robustness. The optimization objective is to minimize the negative log-likelihood loss while maximizing the entropy with weight $\lambda$:

$$\min_\vartheta \max_\lambda \quad \mathbb{E}_{h_t \sim \mathcal{D}_e}[-\log \pi_\vartheta(a_t \mid h_t) - \lambda \mathcal{H}[\pi_\vartheta(\cdot \mid h_t)]] \tag{7}$$

where $a_t$ and $h_t$ are sampled from $D_e$, and the policy $\pi_\vartheta(\cdot \mid h_t) = \tilde{a}_t = \mathcal{N}(\mu_\vartheta(h_t), \Sigma_\vartheta(h_t))$ adopts the stochastic Gaussian policy representation and $\vartheta$ is the policy parameter.

**Generative World Model.** To predict states and rewards, we use $x_t = \{h_t \cup a_t\} = \{\hat{R}_{-K:t}, s_{-K:t}, a_{-K:t}\}$ as the input, which is encoded by linear layers and passes through the casual

transformer with two additional decoded layers to predict the current reward $\tilde{r}_{t-1}$ and the next state $\tilde{s}_t$. The optimization objective for the generative world model is to minimize the mean squared error for the current reward and next state, defined as:

$$\min_{\varphi,\mu} \mathbb{E}[(r_t - L_\varphi^{\tilde{r}}(x_t))^2 + (s_{t+1} - L_\mu^{\tilde{s}}(x_t))^2] \tag{8}$$

where $L_\varphi^{\tilde{r}}$ and $L_\mu^{\tilde{s}}$ are the reward and state generation network for the generative world model, with parameters $\varphi$ and $\mu$.

In the model-based offline RL framework, the policy model and the generative world model have the objectives of generating actions, rewards, and the next state, respectively. The causal transformer structure is used to extract historical information for both the policy and the generative world models. During training, the causal transformer is trained alongside the above models, with the goal of simultaneously minimizing Equations 7 and 8 until the convergence of the models.

**Generating Violating Data.** In RL, excessively high rewards, surpassing those provided by domain experts, may incentivize agents to violate the constraints to maximize the total reward (Liu et al., 2022; 2023). Therefore, we set a high initial target reward $\hat{R}_1$ to obtain violation data. We feed $\hat{R}_1$ and initial state $s_1^{(i)}$ into the model-based offline RL to generate $\tau_v^{(i)}$ in an auto-regressive manner, as depicted in model-based offline RL of Figure 1, where $\tilde{a}, \tilde{r}$ and $\tilde{s}$ are predicted by the model. The target reward $\hat{R}$ decreases incrementally and can be represented as $\hat{R}_{t+1} = \hat{R}_t - \tilde{r}_t$. Considering the average error in trajectory prediction, we generate trajectories with the length $K = 10$. Repeating $N$ initial states, we can get violating data $\mathcal{D}_v = \{\tau_v^{(i)}\}_{i=1}^N$. The detailed parameter settings and sensitivity analysis can be found in Appendix D.

Note that certain other generative models, such as Variational Auto-Encoder (VAE) (Kim et al., 2021), Generative Adversarial Networks (GAN) (Hsu et al., 2021; Iyer et al., 2019), and Denoising Diffusion Probabilistic Models (DDPM) (Croitoru et al., 2023), may be better at generating data. We introduce the model-based offline RL primarily because it has been shown to generate violating data with exploration (Liu et al., 2023) and possess the ability to process time-series features efficiently.

### 4.3 SAFE-CRITICAL DECISION MAKING WITH CONSTRAINTS

To train offline CT, we gather the medical expert dataset $\mathcal{D}_e$ from the environment. Then, we employ gradient descent to train the model-based offline RL, guided by Equation 7 and Equation 8, continuing until the model converges. Using this RL model, we automatically generate violating data denoted as $\mathcal{D}_v$. Subsequently, CT is optimized based on Equation 6 to get the cost function $C$, leveraging samples from both $\mathcal{D}_e$ and $\mathcal{D}_v$. To learn a safe policy, we train the policy $\pi$ using $C$ until it converges based on Equation 1. The detailed training procedure is presented in Algorithm 1.

---

**Algorithm 1** Safe Policy Learning with Offline CT

---

**Input:** Expert trajectories $\mathcal{D}_e$, context length $K$, target reward $\hat{R}_1$, samples $N$, episode length $T$
1: Train model-based offline RL $\mathcal{M}$: Update $\vartheta$, $\varphi$ and $\mu$ using the Equation (7) and Equation (8)
2: **for** t = 1,...,T **do**
3:     Sample initial states $S_1$ from $\mathcal{D}_e$
4:     Generate the violating dataset: $\mathcal{D}_v \leftarrow \mathcal{M}.\text{generate\_data}(S_1, \hat{R}_1, K)$
5:     Sample set of trajectories $\{\tau_e^{(i)}\}_{i=1}^N$ and $\{\tau_v^{(i)}\}_{i=1}^N$ from $\mathcal{D}_e$ and $\mathcal{D}_v$
6:     Train offline CT: Use $\{\tau_e^{(i)}\}_{i=1}^N$ and $\{\tau_v^{(i)}\}_{i=1}^N$ to update $\phi$ based on Equation (6)
7:     Safe policy learning: Update $\pi$ using the cost function $C_\phi(\tau)$ based on Equation (1)
8: **end for**
**Output:** $\pi$ and $C(\tau)$

---

## 5 EXPERIMENT

In this section, we first provide a brief overview of the task, as well as data extraction and preprocessing. Subsequently, in Section 5.1, we demonstrate that CT can describe constraints in healthcare and capture critical patient states. We emphasize its applicability to various CRL methods and its ability to align the expert policy for reducing mortality rates in Section 5.2. Section 5.3 discusses the realization of the objective of safe medical policies. Finally, we use offline policy evaluation (OPE) methods to estimate our policy in the field of dynamic treatment regimes in Section E.2.2.

**Tasks.** We primarily use the sepsis task that is commonly used in previous works (Huang et al., 2022; Raghu et al., 2017a; Komorowski et al., 2018; Do et al., 2020), and supplement some experiments on the mechanical ventilator task (Kondrup et al., 2023; Peine et al., 2021). The detailed definition of the two tasks mentioned above can be found in Appendix A.1 and A.2. For detailed experiments on the mechanical ventilator task, please refer to Appendix E.2.

**Data Extraction and Pre-processing.** Our medical dataset is derived from the Medical Information Mart for Intensive Care III (MIMIC-III) database (Johnson et al., 2016). For each patient, we gather relevant physiological parameters, including demographics, lab values, vital signs, and intake/output events. Data is grouped into 4-hour windows, with each window representing a time step. In cases of multiple data points within a step, we record either the average or the sum. We eliminate variables with significant missing values and use the $k$-nearest neighbors method to fill in the rest.

**Model-based Offline RL Evaluation.** To ensure the rigor of the experiments, we evaluate the validity of the model-based offline RL, as detailed in Appendix D.

### 5.1 CAN OFFLINE CT LEARN EFFECTIVE CONSTRAINTS?

In this section, we primarily assess the efficacy of the cost function learned by offline CT in sepsis, focusing on its capability to evaluate patient mortality rates and capture critical events. First, we employ the cost function to compute cost values for the validation dataset. Subsequently, we statistically analyze the relationship between these cost values and mortality rates. As shown in Figure 4, there is an increase in patient mortality rates with rising cost values. It is noteworthy that such increases in mortality rates are often attributed to suboptimal medical decisions. Therefore, these experimental findings affirm that the cost values effectively reflect the quality of medical decision-making. To observe the impact of the atten-

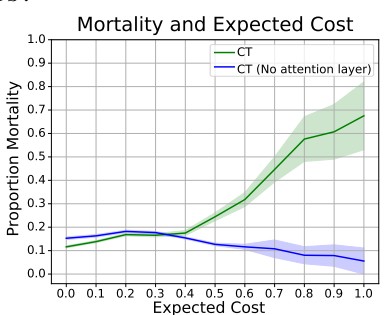

Figure 4: The relationship between cost and mortality.

tion layer (Non-Markovian layer), we conduct experiments by removing the attention layer from CT. The results reveal that the penalty values do not correlate proportionally with mortality rates (shown in Figure 4). This indicates that the attention layer plays a crucial role in assessing constraints.

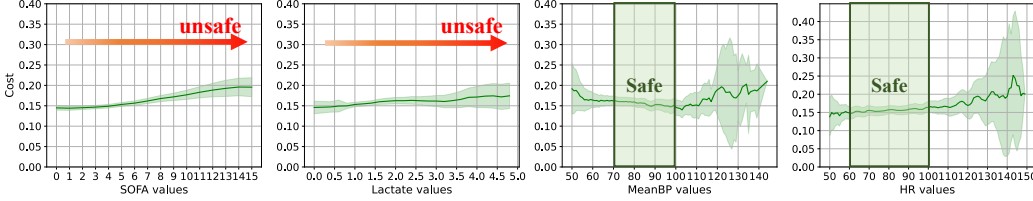

Figure 5: The relationship between physiological indicators and cost values. As SOFA and lactate levels become increasingly unsafe, the cost increases. Mean BP and HR at lower values within the safe range incur a lower cost, but as they move into unsafe ranges, the cost increases, penalizing previous state-action pairs. The cost can differentiate between relatively safe and unsafe regions.

To assess the capability of the cost function to capture key events, we analyze the relationship between physiological indicators and cost values. We focus on four key indicators in sepsis treatment: Sequential Organ Failure Assessment (SOFA) score (Li et al., 2020), lactate levels (Ryoo et al., 2018), Mean Arterial Pressure (MeanBP) (Pandey et al., 2014), and Heart Rate (HR) (Carrara et al., 2018). The SOFA score and lactate levels are critical indicators for assessing sepsis severity, with higher values indicating greater patient risk. MeanBP and HR are essential physiological metrics, typically ranging from 70 to 100 mmHg and 60 to 100 beats, respectively. Deviations from these ranges can signify patient risk. As depicted in Figure 5, the cost values effectively distinguish between high-risk and safe conditions, reflecting changes in patient status. Moreover, we demonstrate that the cost function can capture the dangerous states of other feature variables, including hidden variables. For more detailed information, refer to Appendix E.2.

### 5.2 CAN OFFLINE CT IMPROVE THE PERFORMANCE OF CRL?

**Baselines.** We adopt the DDPG method as the baseline in sepsis research (Huang et al., 2022). Since no other offline inverse reinforcement learning works are available for reference, we have included

three additional settings: no cost, custom cost, and LLMs cost. In the case of no cost, the cost is set to zero, while the design of custom constraints and LLMs cost are outlined in Appendix C. These settings help evaluate whether CT can infer effective constraints.

**Metrics.** To assess effectiveness, we use $\omega$ to indicate the aligning rate with the expert policy and analyze the relationship between DIFF and mortality rate through a graph.

**Results.** We combine our method CT with CRL algorithms (e.g., VOCE, COpiDICE, BCQ-Lag, and CDT), and compare them with both no-cost and custom cost settings. Each CRL model is trained using no cost, custom cost, and CT separately, with other parameters set the same during training. For evaluation metrics, we use IV difference (IV DIFF), vaso difference (VASO DIFF), and combined [IV, VASO] difference (ACTION DIFF) as the metrics to be ranked. We measure the mean and variance of $\omega\%$ in 10 sets of random seeds, and the results are shown in Table 2. From the results, we can conclude: 1) CT makes the strategies in the VOCE, CopiDICE, and CDT methods closer to the lower mortality strategies. 2) CDT+CT achieves better results on all three metrics. CDT is also a transformer-based method, which indicates that transformer-based architecture indeed exhibits more outstanding performance in healthcare.

Figure 6 shows the relationship between IV and VASO DIFF with mortality rates under the DDPG and CDT+CT methods in sepsis. In VASO DIFF, when the gap is zero, the mortality rate under CDT+CT is lower than that under DDPG, indicating that following the former strategy could lead to a lower mortality rate. Similarly, in IV DIFF, the same trend is observed. Notably, for the IV strategy, the lowest mortality rate for DDPG does not occur at the point where the difference is zero, indicating a significant estimation bias.

Table 2: Performance of sepsis strategies under various offline CRL models and different constraints.

| Model | Cost | $\omega_{\text{IV DIFF}}\%\uparrow$ | $\omega_{\text{VASO DIFF}}\%\uparrow$ | $\omega_{\text{ACTION DIFF}}\%\uparrow$ |
|---|---|---|---|---|
| DDPG | - | 50.95±1.34 | 51.45±0.75 | 51.15±1.15 |
| VOCE | No cost | 47.45±0.52 | 46.35±1.82 | 51.00±0.86 |
| | Custom cost | 46.45±0.46 | 52.00±0.98 | 49.40±1.04 |
| | LLMs cost | 48.15±1.23 | 48.90±0.77 | 50.70±1.68 |
| | CT | **53.33±0.94** | **59.04±1.13** | **56.15±1.08** |
| CopiDICE | No cost | 48.30±0.91 | 60.10±0.60 | 51.25±0.70 |
| | Custom cost | **53.05±1.35** | 55.20±0.24 | 53.90±1.04 |
| | LLMs cost | 51.05±1.50 | 58.95±0.38 | 54.35±0.89 |
| | CT | 51.95±0.41 | **60.85±1.08** | **54.60±0.60** |
| BCQ-Lag | No cost | 47.50±1.32 | 51.05±0.61 | 49.35±1.08 |
| | Custom cost | 51.54±0.16 | **56.23±1.43** | 53.69±1.62 |
| | LLMs cost | **56.44±0.75** | 53.59±1.15 | **57.88±0.72** |
| | CT | 52.45±1.01 | 55.34±1.20 | 54.49±0.86 |
| CDT | No cost | 56.50±0.81 | 62.45±1.20 | 58.90±1.34 |
| | Custom cost | 54.70±1.12 | 59.85±1.51 | 57.80±1.00 |
| | LLMs cost | 52.45±0.80 | 60.15±1.17 | 56.35±1.59 |
| | CT | **57.15±1.67** | **65.20±1.22** | **60.00±1.49** |
| CDT No attention layer | | 56.70±0.64 | 62.50±1.57 | 59.10±0.44 |
| Generative Model | - | 55.49±2.55 | 56.60±1.33 | 57.00±2.06 |

**Blue:** Safe policy has a higher aligning rate with the expert policy.
**Bold:** The better cost in each CRL model. ↑: higher is better.

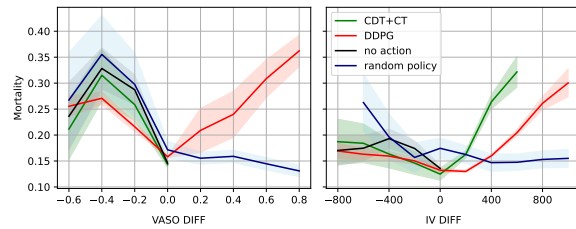

Figure 6: The relationship between DIFF and the mortality rate. The x-axis represents the DIFF. The y-axis indicates the mortality rate of patients at a given DIFF. The solid line represents the mean, while the shaded area indicates the Standard Error of the Mean (SEM).

### 5.3 CAN CRL WITH OFFLINE CT LEARN SAFE POLICIES?

We have confirmed the existence of two unsafe strategy issues, namely "too high" and "sudden change" in the treatment of sepsis, particularly in vaso in Section 1. To validate whether the CRL+CT approach could address these concerns, we employ

Table 3: Proportion of unsafe actions recommended by policies.

| Drug dosage $(\mu g/(kg \cdot min))$ | Physician | DDPG | CDT No cost | CDT Custom cost | CDT CT |
|---|---|---|---|---|---|
| vaso >0.75 | 2.27% | 7.44% | 0.13% | 0%↓ | 0%↓ |
| vaso >0.9 | 1.71% | 7.40% | 0.09% | (max = 0.00) | (max = 0.11) |
| $\Delta$ vaso >0.75 | 2.45% | 21.00% | 0.64% | 0%↓ | 0%↓ |
| $\Delta$ vaso >0.9 | 1.88% | 20.62% | 0.48% | (max $\Delta$ = 0.00) | (max $\Delta$ = 0.10) |

↓: lower is better. max: the maximum drug dosage.
max $\Delta$: the maximum change in drug dosage.

the same statistical methods to evaluate our methodology, shown in Table 3. To elucidate the efficacy of CT, we compare it with CDT+No-cost and CDT+Custom-cost approaches. We find that only the custom cost and CT methods successfully mitigated these unsafe behaviors. However, the custom cost mitigates risks by avoiding medication ($max = 0$). This is an overly conservative strategy. The CDT+CT approach can give a more appropriate drug dosage and is not an overly conservative strategy. In addition, there may be other safety issues that we have not yet verified. Future testing can be conducted on simulation systems such as DTR-Bench (Luo et al., 2024b), detailed in Appendix G.

## 5.4 OFF-POLICY EVALUATION

**Baselines.** 1) Naive baselines. A naive baseline can provide worst-case scenario benchmarks for algorithm evaluation (Luo et al., 2024a), including random policy $\pi_r$, zero-drug policy $\pi_{min}$, max-drug policy $\pi_{max}$, alternating policy $\pi_{alt}$, and weight policy $\pi_{weight}$. 2) RL methods baselines. We select common RL methods such as Deep Q-Network (DQN), Conservative Q-Learning (CQL), Implicit Q-Learning (IQL), Batch-constrained Q-Learning (BCQ), and TD3+BC as baseline models. 3) Cost baselines. We use the no-cost and custom-cost CDT methods as the comparison baselines.

**Metrics.** A recent series of studies have applied offline policy evaluation techniques to dynamic treatment regimes, such as Weighted Importance Sampling (WIS) (Kidambi et al., 2020; Nambiar et al., 2023). To evaluate the policy more accurately, we use metrics such as RMSE and F1 score to describe the deviation from the clinician's policy.

We used the same reward function to compare the policy results under different evaluation metrics, as shown in Table 4. Our findings present that the CDT+CT method outperforms other methods in terms of $RMSE_{IV}$, WIS, $WIS_b$, and $WIS_{bt}$ evaluation metrics. We recognize that safer and more conservative policies may prioritize optimizing safety and constraint compliance, rather than minimizing the statistical difference from clinician behavior. As a result, this policy may lead to poorer performance of the model on metrics such as F1 and $RMSE_{VASO}$.

Table 4: Comparison across policies on the sepsis test set. The best algorithms are highlighted in red. $RMSE_{IV}$ and $RMSE_{VASO}$ mean the RMSE loss for the IV fluid treatment and vasopressor treatment. P.F1 and S.F1 denote the patient-wise F1 and sample-wise F1.

| Metric | alt | max | min | random | weight | DQN | CQL | IQL | BCQ | TD3+BC | CDT (No cost) | CDT (Custom cost) | CDT+CT |
|---|---|---|---|---|---|---|---|---|---|---|---|---|---|
| $RMSE_{IV}$ ↓ | 763.89 | 861.51 | 645.83 | 671.39 | 645.83 | 638.51 ± 8.63 | 541.67 ± 5.74 | 578.96 ± 10.06 | 626.2 ± 9.56 | 978.83 ± 35.62 | 435.89 ± 19.60 | 484.51 | 433.55 ± 7.20 |
| $RMSE_{VASO}$ ↓ | 0.67 | 0.89 | 0.32 | 0.5 | 0.59 | 0.44 ± 0.07 | 0.30 ± 0.01 | 0.31 ± 0.01 | 0.31 | 1.61 | 1.14 | 1.16 | 1.13 ± 0.01 |
| WIS ↑ | −4.58 | −4.62 | −4.58 | −3.84 | −3.78 | −3.79 ± 0.01 | −4.10 ± 1.43 | −5.83 | −5.14 ± 1.36 | −4.58 | −5.75 ± 2.13 | −5.13 | −3.51 ± 0.11 |
| $WIS_b$ ↑ | −5.43 | −4.81 | −5.76 | −4.4 | −4.73 | −3.88 ± 0.73 | −4.48 ± 0.77 | −5.31 ± 0.06 | −5.41 ± 0.17 | −4.95 ± 0.19 | −5.38 ± 1.73 | −4.59 ± 0.14 | −3.52 ± 0.17 |
| $WIS_t$ ↑ | −4.58 | −4.62 | −4.58 | −3.97 | −3.78 | −3.84 ± 0.11 | −4.10 ± 1.43 | −5.83 | −5.14 ± 1.36 | −4.58 | −5.75 ± 2.13 | −5.13 | −3.51 ± 0.11 |
| $WIS_{bt}$ ↑ | −5.64 | −4.69 | −5.61 | −4.5 | −4.5 | −3.87 ± 0.67 | −4.38 ± 0.98 | −5.27 ± 0.05 | −5.55 ± 0.19 | −4.90 ± 0.10 | −5.27 ± 1.72 | −4.68 ± 0.10 | −3.52 ± 0.17 |
| P.F1 ↑ | 0.2 | 0.02 | 0.2 | 0.2 | 0.0 | 0.06 ± 0.02 | 0.33 ± 0.01 | 0.34 ± 0.01 | 0.23 ± 0.01 | 0.02 ± 5.98 | 0.19 ± 0.04 | 0.18 | 0.17 ± 0.02 |
| S.F1 ↑ | 0.19 | 0.02 | 0.19 | 0.19 | 0.0 | 0.06 ± 0.02 | 0.32 ± 0.01 | 0.33 ± 0.01 | 0.22 ± 0.01 | 0.02 | 0.18 ± 0.04 | 0.17 | 0.16 ± 0.02 |

↓: lower is better. ↑: higher is better. Red: It highlights which method in the corresponding row performs better on the given metric.
$WIS_b$, $WIS_t$ and $WIS_{bt}$: WIS methods are optimized for variance reduction through bootstrapping, ratio truncation and a combination of both.

**Ablation Study.** To investigate the impact of each component on the model's performance, we conducted experiments by sequentially removing each component from the CDT+CT model. The results are presented in the lower half of Table 2. Both CT and its Non-Markovian layer (attention layer) are essential components; removing either one results in a decrease in performance. Additionally, we observed that even a pure generative model outperforms DDPG in terms of performance. This is primarily because it inherently operates as a sequence-based reinforcement learning model, possessing exploration and consideration for long-term history. Therefore, this further underscores the effectiveness of sequence-based approaches in healthcare applications. To further analyze the performance of different sequence models, we conduct offline policy evaluation on models based on LSTM and transformer architectures. We found that the latter performs better, see Appendix E.3.

# 6 CONCLUSION

In this paper, we propose offline CT, a novel ICRL algorithm designed to address safety issues in healthcare. This method utilizes a causal attention mechanism to observe patients' historical information, similar to the approach taken by actual doctors, and employs Non-Markovian importance weights to effectively capture critical states. To achieve offline learning, we introduce a model-based offline RL for exploratory data augmentation to discover unsafe decisions and train CT. Experiments in sepsis and mechanical ventilation demonstrate that our method avoids risky behaviors while achieving strategies that closely approximate the lowest mortality rates.

**Limitations.** There are also several limitations of offline CT: 1) Lack of rigorous theoretical analysis. We did not precisely define the types of constraint sets, thereby conducting rigorous theoretical analysis on constraint sets remains challenging. 2) Need for more computational resources. Due to the Transformer architecture, more computational resources are required. 3) Unrealistic assumptions of expert demonstrations. We assume that the policy capable of treating patients to survival is the expert policy. However, in reality, this assumption may not always hold. Therefore, researching a more effective approach to address the aforementioned issues holds promise for the field of secure medical reinforcement learning.

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

# A PROBLEM DEFINE

## A.1 SEPSIS PROBLEM DEFINE

Our definition is similar to (Raghu et al., 2017b). We extract data from adult patients meeting the criteria for sepsis-3 criteria (Singer et al., 2016) and collect their data within the first 72 hours of admission.

**State Space.** We use a 4-hour window and select 48 patient indicators as the state for a one-time unit of the patient. The state indicators include Demographics/Static, Lab Values, Vital Signs, and Intake and Output Events, detailed as follows (Raghu et al., 2017b):

- Demographics/Static: Shock Index, Elixhauser, SIRS, Gender, Re-admission, GCS - Glasgow Coma Scale, SOFA - Sequential Organ Failure Assessment, Age
- Lab Values Albumin: Arterial pH, Calcium, Glucose, Hemoglobin, Magnesium, PTT - Partial Thromboplastin Time, Potassium, SGPT - Serum Glutamic-Pyruvic Transaminase, Arterial Blood Gas, BUN Blood Urea Nitrogen, Chloride, Bicarbonate, INR - International Normalized Ratio, Sodium, Arterial Lactate, CO2, Creatinine, Ionised Calcium, PT - Prothrombin Time, Platelets Count, SGOT Serum Glutamic-Oxaloacetic Transaminase, Total bilirubin, White Blood Cell Count
- Vital Signs: Diastolic Blood Pressure, Systolic Blood Pressure, Mean Blood Pressure, PaCO2, PaO2, FiO2, PaO/FiO2 ratio, Respiratory Rate, Temperature (Celsius), Weight (kg), Heart Rate, SpO2
- Intake and Output Events: Fluid Output - 4 hourly period, Total Fluid Output, Mechanical Ventilation

**Action Space.** Regarding the treatment of sepsis, there are two main types of medications: intravenous fluids and vasopressors. We select the total amount of intravenous fluids for each time unit and the maximum dose of vasopressors as the two dimensions of the action space, defined as $(\mathrm{sum}(\mathrm{IV}), \max(\mathrm{Vaso}))$. Each dimension is a continuous value greater than $0$. The data distribution of the doctor's actions is shown in Figure. 7.

**Reward Function.** We refer to the reward function used in (Huang et al., 2022), as shown in the following equation:

$$r\left(s_t, s_{t+1}\right) = \lambda_1 \tanh\left(s_t^{\mathrm{SOFA}} - 6\right) + \lambda_2\left(s_{t+1}^{\mathrm{SOFA}} - s_t^{\mathrm{SOFA}}\right)) \tag{9}$$

Where $\lambda_0$ and $\lambda_1$ are hyperparameters set to $-0.25$ and $-0.2$, respectively. This reward function is designed based on the SOFA score, as it is a key indicator of the health status of sepsis patients and is widely used in clinical settings. The formula describes a penalty when the SOFA score increases and a reward when the SOFA score decreases. We set 6 as the cutoff value because the mortality rate sharply increases when the SOFA score exceeds 6 (Ferreira et al., 2001).

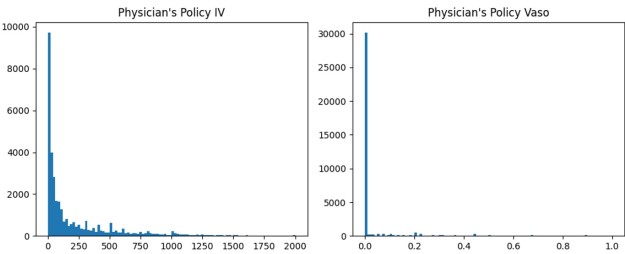

Figure 7: Actions distribution under physician's policy in sepsis.

## A.2 MECHANICAL VENTILATION TREATMENT PROBLEM DEFINE

The RL problem definition for Mechanical Ventilation Treatment is referenced from (Kondrup et al., 2023).

**State Space.** We also use a 4-hour window and select 48 patient indicators as the state for a one-time unit of the patient. The state indicators are as follows:

- Demographics/Static: Elixhauser, SIRS, Gender, Re-admission, GCS, SOFA, Age
- Lab Values Albumin: Arterial pH, Glucose, Hemoglobin, Magnesium, PTT, BUN Blood Urea Nitrogen, Chloride, Bicarbonate, INR, Sodium, Arterial Lactate, CO2, Creatinine, Ionised Calcium, PT, Platelets Count, White Blood Cell Count, Hb
- Vital Signs: Diastolic Blood Pressure, Systolic Blood Pressure, Mean Blood Pressure, Temperature, Weight (kg), Heart Rate, SpO2
- Intake and Output Events: Urine output, vasopressors, intravenous fluids, cumulative fluid balance

**Action Space.** The action space mainly consists of Positive End Expiratory Pressure (PEEP) and Fraction of Inspired Oxygen (FiO2), which are crucial parameters in ventilator settings. Here, we consider a discrete space configuration, with each parameter divided into 7 intervals. Therefore, our action space is $7 \times 7$, depicted as 5.The data distribution of the doctor's actions is shown in Figure.8.

Table 5: The action space of the mechanical ventilator.

| Action | 0 | 1 | 2 | 3 | 4 | 5 | 6 |
|---|---|---|---|---|---|---|---|
| PEEP($cmH_2 0$) | 0-5 | 5-7 | 7-9 | 9-11 | 11-13 | 13-15 | >15 |
| FiO2(Percentage(%)) | 25-30 | 30-35 | 35-40 | 40-45 | 45-50 | 50-55 | >55 |

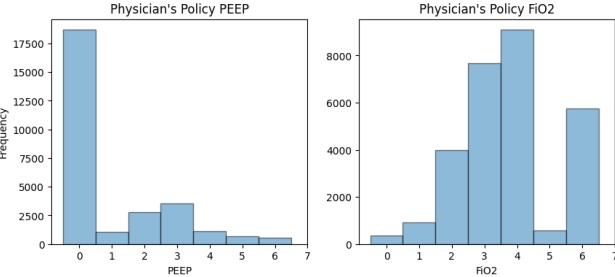

Figure 8: Actions distribution under physician's policy in mechanical ventilation.

**Reward Function.** The primary objective of setting respiratory parameters is to ensure the patient's survival. We adopt the same reward function design as the work (Kondrup et al., 2023), defined as Equation 10. This reward function first considers the terminal reward: if the patient dies, the reward $r$ is set to $-1$; otherwise, it is $+1$ in the terminal state. Additionally, to provide more frequent rewards, intermediate rewards are considered. Intermediate rewards mainly focus on the Apache II score, which evaluates various parameters to describe the patient's health status. This reward function utilizes the increase or decrease in this score to reward the agent.

$$r\left(s_t, a_t, s_{t+1}\right) = \begin{cases} +1 & \text{if } t = T \text{ and } m_t = 1 \\ -1 & \text{if } t = T \text{ and } m_t = 0 \\ \frac{(A_{t+1} - A_t)}{\max_A - \min_A} & \text{otherwise} \end{cases} \quad (10)$$

In Equation 10, $T$ represents the length of the patient's trajectory, $m$ indicates whether the patient ultimately dies, $A$ denotes the Apache II score, and $\max_A$ and $\min_A$ respectively denote the maximum and minimum values.

## B  UNSAFE BEHAVIOR ANALYSIS

We conducted additional experiments by stratifying patients based on their SOFA scores into three categories: mild, moderate, and severe sepsis, shown in Figure. 9. Our findings reveal that the DDPG model tends to overestimate medication dosages for patients with mild and moderate sepsis. These patients, in many cases, do not require such high dosages.

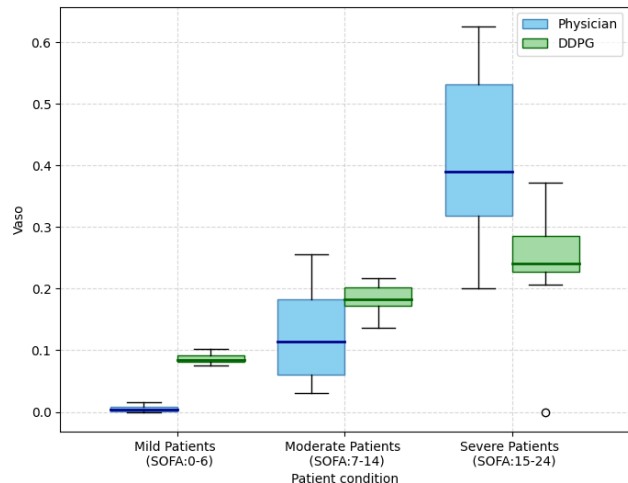

Figure 9: The distribution of medication doses across different patient condition in sepsis.

## C DESIGN AND ANALYSIS OF THE CUSTOM AND LLMs COST FUNCTION

### C.1 CUSTOM COST FUNCTION

We base our design on prior knowledge that intravenous (IV) intake exceeding $2000mL/4h$ or vaso-pressor (Vaso) dosage surpassing $1g/(kg \cdot min)$ is generally considered unsafe in sepsis treatment (Shi et al., 2020). To design a reasonable constraint function, we refer to the constraint function designed by Liu *et al.* in the Bullet safety gym environments (Liu et al., 2023). We define the cost function as shown in Equation 11. Thus, during the treatment of sepsis, if the agent exceeds the maximum dosage thresholds of the two medications, it incurs a cost due to constraint violation.

$$c(s, a) = \mathbf{1}(a_{\text{IV}} > a_{\text{IV}_{\max}}) + \mathbf{1}(a_{\text{Vaso}} > a_{\text{Vaso}_{\max}}) \tag{11}$$

where, $s$ and $a$ represent the patient's state and action, respectively. $a_{IV\ \max} = 2000$ indicates that the maximum fluid intake through IV is $2000mL$, and $a_{Vaso\ \max} = 1$ signifies that the maximum Vaso dosage is $1\mu g/(kg \cdot min)$.

We applied our custom constraint function in the CDT (Liu et al., 2023) method, and the results are shown in Figure 10. Compared to the Vaso dosage recommended by doctors, our strategy exhibits excessive suppression of the Vaso. The maximum dosage of Vaso is $0.0011\mu g/(kg \cdot min)$, which is minimal and insufficient to provide the patient with effective therapeutic effects.

Therefore, Equation 11 is not suitable. The primary issues may include uniform constraint strength for excessive drug dosages, for instance, the cost for IV exceeding 2000 mL and IV exceeding 3000 mL is the same at 1; lack of generalization, where the constraint cost does not vary with the patient's tolerance. If a patient has an intolerance to VASO, the maximum value for VASO maybe 0, which cannot be captured by the self-imposed constraint function. Moreover, it lacks generalization, requiring redesign of the constraint function when addressing other unsafe medical issues; and it's essential to ensure the correctness of the underlying medical knowledge premises.

### C.2 LLMs COST FUNCTION

We provide prior knowledge to GPT-4.0, and the cost function it designs is as shown in Equation 12. Based on the self-designed constraint function, LLMs added a penalty for Vaso doses mutations, giving the agent a certain penalty when the change in Vaso doses exceeds the threshold.

$$c(s, a) = \mathbf{1}(a_{\text{IV}} > a_{\text{IV}_{\max}}) + \mathbf{1}(a_{\text{Vaso}} > a_{\text{Vaso}_{\max}}) + \mathbf{1}((a_{\text{vaso}} - a_{\text{Vaso}_{\text{prev}}}) > a_{\text{vaso}_{\text{change.threshold}}}) \tag{12}$$

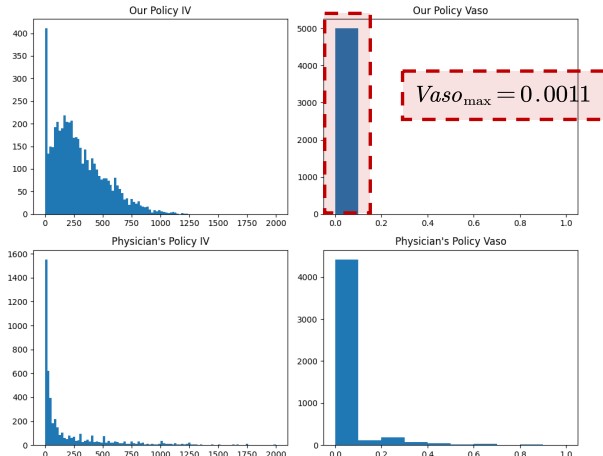

Figure 10: Drug dosage distribution under custom constraint functions in sepsis.

# D    THE EVALUATION OF MODEL-BASED OFFLINE RL

## D.1    THE LENGTH OF A TRAJECTORY.

Regarding the selection of trajectory length, we consider the relationship between the average prediction error, the error of the last point in the trajectory, and the trajectory length. We use the model-based offline RL to generate trajectories and compare them with expert data using the Euclidean distance to measure their differences. We evaluate the average error and the error of the last point in the trajectory, as shown in Figure 11. We observe that with an increase in trajectory length, the average prediction error at each time step decreases, while the state error stabilizes. Taking into account the observation length and prediction accuracy, we ultimately choose to generate trajectories with lengths ranging from 10 to 15.

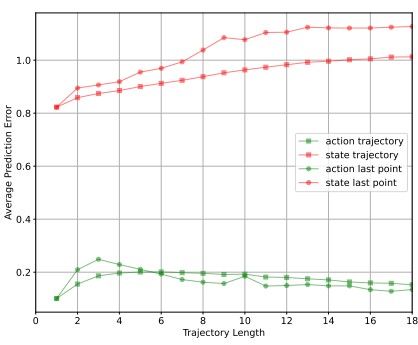

Figure 11: The relationship between average prediction error and trajectory length.

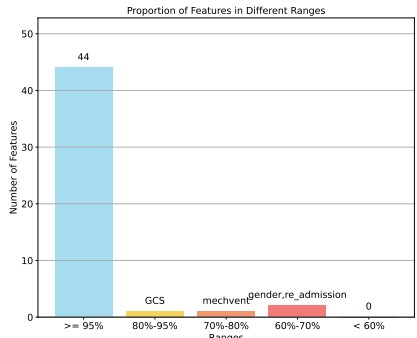

Figure 12: The accuracy of predicting different state values within the legal range.

## D.2    GENERATING DATA WITHIN A REASONABLE RANGE.

To validate model-based offline RL, we first check whether the values it produces fall within the legal range. The results are depicted in Figure 12. After analyzing the generated data, we find that the majority of state values have a probability of over 99% of being within the legal range. A few values related to gender and re-admission range between 60% and 70%. This could be due to these two indicators having limited correlation with other metrics, making them more challenging for the model to assess.

## D.3 Generating violating data.

In addition, we evaluate the violating actions generated by the model, as shown in Figure 13. When compared with expert strategies and penalty distributions, we find that the actions generated by the model mostly fall within the legal range. However, it occasionally produces behaviors that are inappropriate for the current state, constituting violating data. This indicates that our generative model can produce legally violating data.

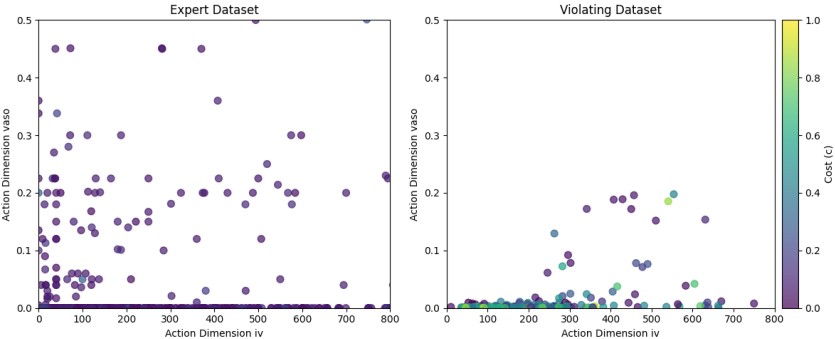

Figure 13: The distribution and penalty values of violating data and expert data.

## D.4 The sensitivity of CT models to generative models and reward setting.

We designed the following experiment to explore the sensitivity of the estimated policy to the generative world model. Since the quality of data generated by the generative world model depends on the target reward, higher target rewards lead the world model to generate more aggressive data to obtain more rewards. We set the target rewards to 1, 5, 10, 40, and 50, and observed the impact of the generated data on the policy, as shown in Table 6. The policy performance improves as the target reward increases, but it reaches an upper bound and does not increase indefinitely.

Table 6: The impact of generative world models with different target rewards on policy estimation.

| Target Reward | IV DIFF | VASO DIFF | ACTION DIFF |
|---|---|---|---|
| 1 | $51.60 \pm 1.78$ | $58.8 \pm 2.74$ | $54.25 \pm 1.79$ |
| 5 | $52.50 \pm 1.46$ | $58.84 \pm 3.24$ | $54.45 \pm 1.65$ |
| 10 | $52.25 \pm 1.33$ | $56.85 \pm 4.20$ | $55.00 \pm 1.80$ |
| 40 | $52.05 \pm 1.30$ | $56.75 \pm 3.13$ | $55.80 \pm 1.76$ |
| 50 | $52.00 \pm 1.31$ | $57.35 \pm 2.09$ | $55.05 \pm 1.93$ |

# E    THE EVALUATION OF COST FUNCTION

## E.1    The Evaluation of Cost function in Sepsis

### E.1.1    Capture unsafe variables

To validate that the CT method captures key states, we conduct statistical analysis on the relationship between state values and penalty values. We collect penalty values under different state values for all patients, and the complete information is shown in Figure 15. We find that the CT method successfully captures unsafe states and imposes higher penalties accordingly. The safe range of state values is shown in Table 7.

### E.1.2    Capture unsafe hidden variables

In a medical context, mortality rates may be influenced by various factors. The dataset often contains numerous unaccounted features (hidden variables), such as epinephrine, dopamine, medical history,

Table 7: State indicators and their normal ranges.

| Indicator | Safe Range | Indicator | Safe Range | Indicator | Safe Range |
|---|---|---|---|---|---|
| Albumin | 3.5∼5.1 | HCO3 | 25∼40 | SGOT | 0∼40 |
| Arterial_BE | -3∼+3 | Glucose | 70∼140 | SGPT | 0∼40 |
| Arterial_lactate | 0.5∼1.7 | HR | 60∼100 | SIRS | ↓ |
| Arterial_PH | 7.35∼7.45 | Hb | 12∼16 | SOFA | ↓ |
| BUN | 7∼22 | INR | 0.8∼1.5 | Shock_Index | ↓ |
| CO2_mEqL | 20∼34 | MeanBP | 70∼100 | Sodium | 135∼145 |
| Calcium | 8.6∼10.6 | PT | 11∼13 | SpO2 | 95∼99 |
| Chloride | 96∼106 | PTT | 23∼37 | SysBP | 90∼139 |
| Creatinine | 0.5∼1.5 | PaO2_FiO2 | 400∼500 | Temp_C | 36.0∼37.0 |
| DiaBP | 60∼89 | Platelets_count | 125∼350 | WBC_count | 4∼10 |
| FiO2 | 0.5∼0.6 | Potassium | 4.1∼5.6 | PaCO2 | 35∼45 |
| GCS | ↑ | RR | 12∼20 | PaO2 | 80∼100 |

↑ indicates higher values are more normal, while ↓ indicates lower values are more normal.
The maximum value for GCS is 15. The minimum value for SIRS, SOFA, and Shock_Index is 0.

and phenotypes. As noted in (Jeter et al., 2019), clinicians typically set a mean arterial pressure (MAP) target (e.g., 65) and administer vasopressors until the patient reaches a safe pressure level. Additionally, Luo et al. (2024a) suggest using the NEWS2 score as evidence for clinical rewards. To validate whether our penalty function captures changes in hidden variables (NEWS and MAP), we conducted supplementary experiments, as shown in Figure 14. When the NEWS score is excessively high, the penalty value increases accordingly; similarly, when MAP falls outside the normal range, the penalty also rises. This indicates that the penalty function successfully captures changes in hidden variables and compensates for the reward function's omission of certain parameter variables. Therefore, we can rely on a simple reward function and use the penalty function to achieve safe policy learning.

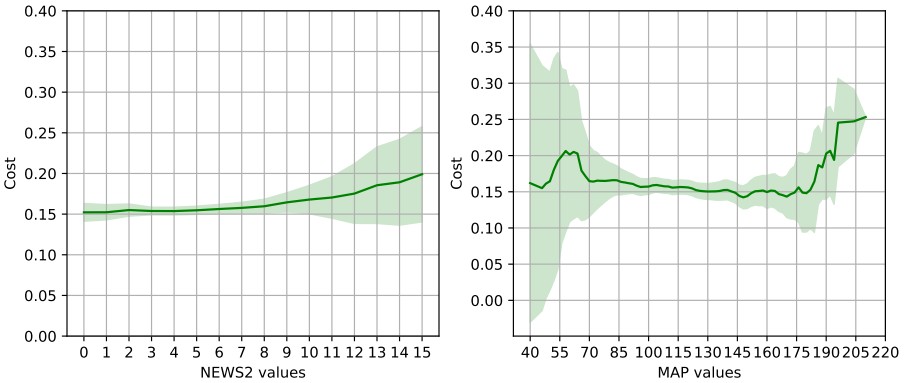

Figure 14: The relationship between the NEWS2 and MAP indicators with cost values.

### E.1.3 ABLATION STUDY: THE ROLE OF THE ATTENTION LAYER.

To validate the role of the attention layer in capturing states in CT, we conducted tests, and the experimental results are presented in Figure 16 and 15. We found that the attention layer plays a crucial role in state capture. For instance, in the case of an increase in the SOFA score, without the attention layer, this increase cannot be captured, while with the attention layer, it clearly captures the change. Thus, this indicates that SOFA, as a key diagnostic indicator of sepsis, with the help of the attention layer, CT can accurately capture its changes.

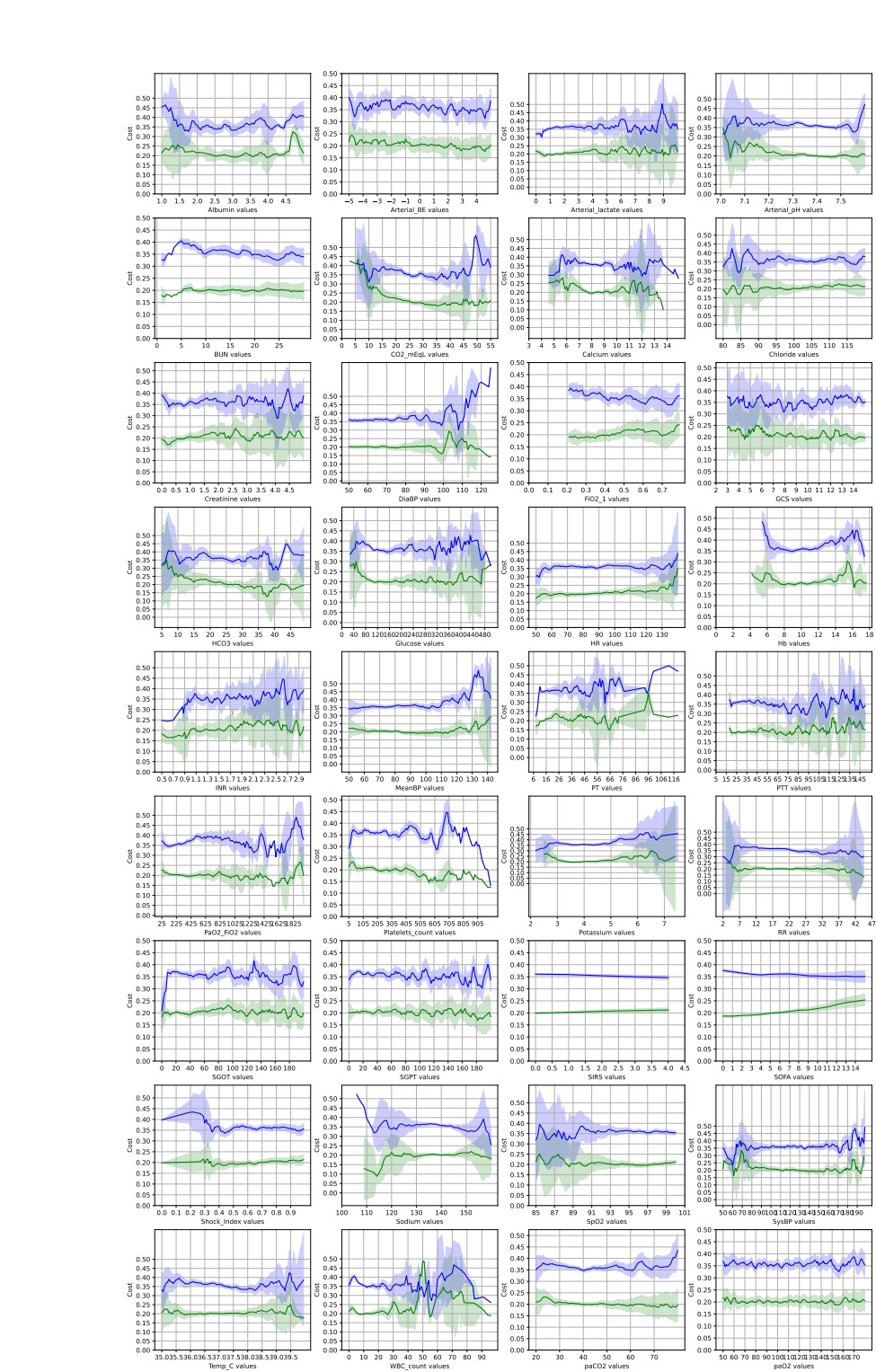

Figure 15: The relationship between all states and cost values

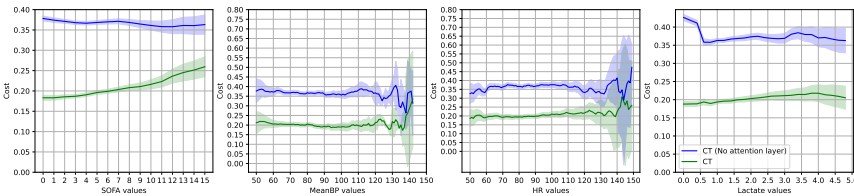

Figure 16: The performance contrast between CT with and without an attention layer. The blue line represents the absence of an attention layer, while the green line indicates the presence of an attention layer.

## E.2 THE EVALUATION OF COST FUNCTION IN MECHANICAL VENTILATOR

### E.2.1 CAN OFFLINE CT IMPROVE THE PERFORMANCE OF CRL?

**Baselines.** We adopt the Double Deep Q-Learning (DDQN) and Conservative Q-Learning (CQL) methods as baselines in ventilator research (Kondrup et al., 2023).

Corresponding experiments are conducted on the mechanical ventilator, as shown in Figure 17. Compared to previous methods DDQN and CQL, under the CDT+CT approach, a noticeable trend is observed where the proportion of mortality rates increases with increasing differences. When there is a significant difference in DIFF, the results may be unreliable, possibly due to the limited data distribution in the tail.

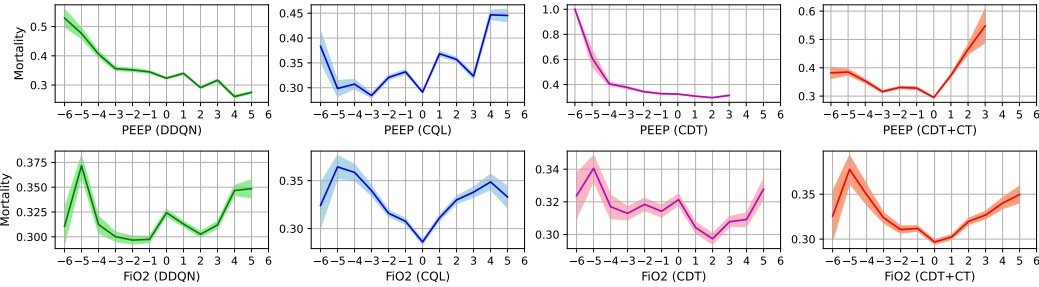

Figure 17: The relationship between the DIFF of actions and mortality in mechanical ventilator. The actions mainly consist of Positive End Expiratory Pressure (PEEP) and Fraction of Inspired Oxygen (FiO2), which are crucial parameters in ventilator settings.

### E.2.2 OFF-POLICY EVALUATION IN MECHANICAL VENTILATOR

**Baselines.** 1) Naive baselines. A naive baseline can provide worst-case scenario benchmarks for algorithm evaluation (Luo et al., 2024a), including random policy $\pi_r$, zero-drug policy $\pi_{\min}$, max-drug policy $\pi_{\max}$, alternating policy $\pi_{\text{alt}}$ and weight policy $\pi_{\text{weight}}$. 2) RL methods baselines. We select common RL methods such as Deep Q-Network (DQN), Conservative Q-Learning (CQL), Implicit Q-Learning (IQL), and Batch Constrained Q-Learning (BCQ) as baseline models.

**Metrics.** A recent series of studies have applied offline policy evaluation techniques to dynamic treatment regimes, including Weighted Importance Sampling (WIS) (Kidambi et al., 2020; Nambiar et al., 2023) and Doubly Robust (DR) estimators (Raghu et al., 2017a; Wu et al., 2023; Wang et al., 2018). To more accurately evaluate the policy, we use metrics such as RMSE and F1 score to describe the deviation from the clinician's policy.

We used the same reward function to compare the policy results under different evaluation metrics in mechanical ventilators, as shown in Table 8. Our findings present that the CDT+CT method outperforms other methods in terms of $\text{RMSE}_{\text{PEEP}}$, WIS, $\text{WIS}_b$, and $\text{WIS}_{\text{bt}}$ evaluation metrics.

Table 8: Comparison across policies on the mechanical ventilator test set. The best algorithms are highlighted in red. $\text{RMSE}_{\text{PEEP}}$ and $\text{RMSE}_{\text{FiO2}}$ mean the RMSE loss for the PEEP and FiO2. P.F1 and S.F1 denote the patient-wise F1 and sample-wise F1.

| Metric | alt | max | min | random | weight | DQN | CQL | IQL | BCQ | CDT+CT |
|---|---|---|---|---|---|---|---|---|---|---|
| $\text{RMSE}_{\text{PEEP}}$ ↓ | 8.15 | 8.96 | 7.30 | 6.01 | 7.16 | $6.15 \pm 0.48$ | $5.51 \pm 0.06$ | $5.51 \pm 0.03$ | $6.15 \pm 0.04$ | $2.88 \pm 0.04$ |
| $\text{RMSE}_{\text{FiO2}}$ ↓ | 21.80 | 14.09 | 27.28 | 18.56 | 26.34 | $15.81 \pm 1.36$ | $13.08 \pm 0.17$ | $13.72 \pm 0.18$ | $16.69 \pm 0.16$ | $13.13 \pm 0.15$ |
| WIS ↑ | 0.66 | 1.01 | 0.66 | 0.66 | 0.84 | $-1.13$ | $0.84 \pm 0.07$ | $0.66 \pm 0.29$ | $0.81 \pm 0.09$ | $0.86 \pm 0.07$ |
| $\text{WIS}_b$ ↑ | 0.16 | 0.7 | 0.14 | 0.7 | 0.83 | $-0.81 \pm 0.13$ | $0.78 \pm 0.07$ | $0.54 \pm 0.23$ | $0.78 \pm 0.06$ | $0.83 \pm 0.10$ |
| $\text{WIS}_t$ ↑ | 0.66 | 1.01 | 0.66 | 0.66 | 0.84 | $-1.13$ | $0.84 \pm 0.07$ | $0.66 \pm 0.29$ | $0.81 \pm 0.09$ | $0.86 \pm 0.07$ |
| $\text{WIS}_{bt}$ ↑ | 0.11 | 0.73 | $-0.02$ | 0.72 | 0.83 | $-0.81 \pm 0.15$ | $0.58 \pm 0.14$ | $0.51 \pm 0.24$ | $0.77 \pm 0.04$ | $0.84 \pm 0.08$ |
| DR ↑ | $-0.14$ | $-0.1$ | $-0.1$ | $-0.47$ | $-0.02$ | $-0.06 \pm 0.02$ | $-0.80 \pm 0.04$ | $-1.30 \pm 0.03$ | $-0.69 \pm 0.05$ | $-0.15 \pm 0.02$ |
| P.F1 ↑ | 0.01 | 0.01 | 0.01 | 0.01 | 0.0 | 0.01 | 0.24 | 0.28 | 0.18 | 0.03 |
| S.F1 ↑ | 0.01 | 0.01 | 0.01 | 0.01 | 0.0 | 0.02 | 0.25 | 0.25 | 0.21 | 0.04 |

↓: lower is better. ↑: higher is better.
$\text{WIS}_b$, $\text{WIS}_t$ and $\text{WIS}_{bt}$: WIS methods are optimized for variance reduction through bootstrapping, ratio truncation and a combination of both.

### E.3 THE EVALUATION OF DIFFERENT SEQUENCE MODELS

To further analyze the performance of different sequence models, we conduct offline policy evaluation on models based on LSTM and transformer architectures. In sepsis and mechanical ventilator environments, the transformer-based models outperform LSTM-based models in a greater number of evaluation metrics, as shown in Table 9.

Table 9: LSTM vs Attention. The best algorithms are highlighted in red.

| Metric | Sepsis | | Mechanical ventilator | |
|---|---|---|---|---|
| | CT(LSTM) | CT(Attention) | CT(LSTM) | CT(Attention) |
| $\text{RMSE}_{\text{action1}}$ ↓ | $505.06 \pm 12.27$ | $433.55 \pm 7.20$ | 8.74 | $2.88 \pm 0.04$ |
| $\text{RMSE}_{\text{action2}}$ ↓ | 1.57 | $1.13 \pm 0.01$ | 19.05 | $13.13 \pm 0.15$ |
| WIS ↑ | $-3.03 \pm 0.31$ | $-3.51 \pm 0.11$ | $-1.05$ | $0.86 \pm 0.07$ |
| $\text{WIS}_b$ ↑ | $-3.55 \pm 0.38$ | $-3.52 \pm 0.17$ | $-0.38 \pm 0.04$ | $0.83 \pm 0.10$ |
| $\text{WIS}_t$ ↑ | $-3.03 \pm 0.31$ | $-3.51 \pm 0.11$ | $-1.05$ | $0.86 \pm 0.07$ |
| $\text{WIS}_{bt}$ ↑ | $-3.64 \pm 0.48$ | $-3.52 \pm 0.17$ | $-0.30 \pm 0.08$ | $0.84 \pm 0.08$ |
| DR ↑ | $-3.05 \pm 0.38$ | $-3.08$ | $-0.13$ | $-0.15 \pm 0.02$ |
| P.F1 ↑ | $0.10 \pm 0.11$ | $0.17 \pm 0.02$ | 0.05 | 0.03 |
| S.F1 ↑ | $0.09 \pm 0.10$ | $0.16 \pm 0.02$ | 0.04 | 0.04 |

## F ANALYSIS OF PATIENT HISTORY COMPRESSION

To evaluate whether the proposed Constraint Transformer (CT) can learn meaningful representations associated with critical safety constraints in the medical domain, we conducted a 2D visualization experiment. The goal was to map patient histories into a two-dimensional space and analyze the separability of "safe" and "unsafe" regions.

**Experimental Setup:**

**Data:** Patient history data included various medical features, such as blood pressure, lactate concentration, and drug dosage. Each patient history comprised a sequence of state-action pairs, aggregated as a feature vector for dimensionality reduction.

**Labeling:** Each point in the 2D space was labeled as "safe" or "unsafe" based on the patient's final state, using different colors (e.g., blue for "safe," red for "unsafe").

**Methods**: Dimensionality reduction methods such as Principal Component Analysis (PCA) Kurita (2019) was applied directly to the raw patient history features.

**Patients' history mapping:** Patients' history were extracted from the output of the CT layer and reduced to two dimensions using PCA.

**CT embedding mapping:** Embeddings $d_t$ were extracted from the output of the CT layer and reduced to two dimensions using PCA.

Dimensionality reduction applied to raw features struggled to capture safety-related information specific to the medical domain. However, the CT layer successfully learned task-specific represen-

tations that reflect critical safety constraints, enabling better discrimination between safe and unsafe patient states, shown in Figure. 18.

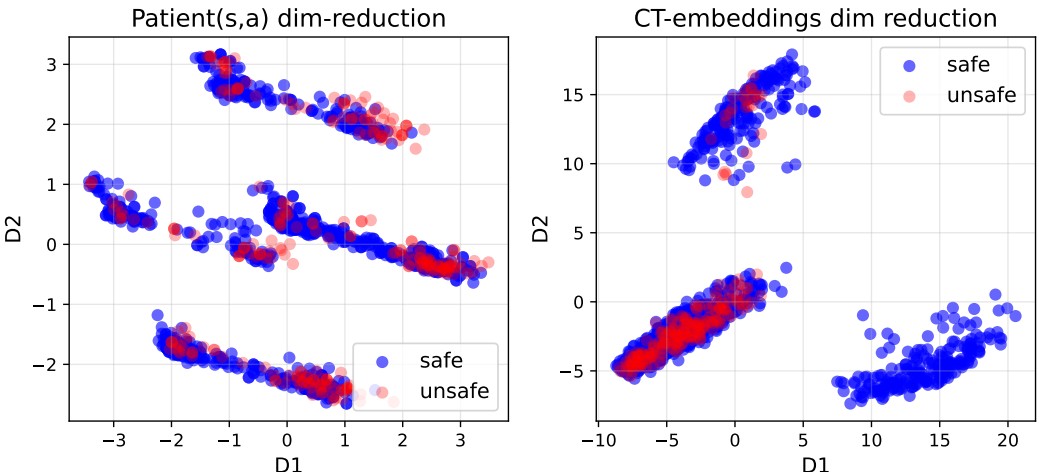

Figure 18: Analysis of Patient History Compression.

## G    ONLINE TESTING METHODS

Currently, some studies (Yoon et al., 2019; Luo et al., 2024b; Brophy et al., 2023) have proposed simulation modeling approaches to address the challenges of directly testing RL-based dynamic treatment regimes in clinical environments. However, since the existing online testing systems (such as DTR-Bench (Luo et al., 2024b)) do not provide expert data in their simulation environment, and the offline method proposed in this paper requires expert datasets to train a safe policy, we are unable to use online testing systems for evaluation. In the future, we can establish an offline testing system to enable the testing of offline reinforcement learning strategies.

## H    EXPERIMENTAL SETTINGS

To train the CRL+CT model, we use a total of 3 NVIDIA GeForce RTX 3090 GPUs, each with 24GB of memory. Training a CRL+CT model typically takes 5-6 hours. We employ 5 random seeds for validation. We use the Adam optimization algorithm to optimize all our networks, updating the learning rate using a decay factor parameterization at each iteration. The main hyperparameters are summarized in Table 10 and 11.

Table 10: List of the utilized hyperparameters in CT.

| Offline CT Parameters | values |
|---|---|
| **Genetivate Model** | |
| Embedding_dim | 128 |
| Layer | 3 |
| Head | 8 |
| Learning rate | 1e-4 |
| Pre-train steps | 5000 |
| Batch size | 256 |
| **CT** | |
| Embedding_dim | 64 |
| Layer | 3 |
| Head | 1 |
| Learning rate | 1e-6 |
| Update steps | 30000 |
| Batch size | 512 |
| **CDT** | |
| Learning rate | 1e-4 |
| Embedding_dim | 128 |
| Layers | 3 |
| Heads | 8 |
| Update steps | 60000 |

Table 11: List of the utilized hyperparameters in CRL.

| Parameters | Sepsis | | Parameters | Mechanical Ventilation |
|---|---|---|---|---|
| **General** | | | **General** | |
| Expert data patient number | 14313 | | Expert data patient number | 13846 |
| Validation data patient number | 6275 | | Validation data patient number | 5954 |
| Max Length | 10 | | Max Length | 10 |
| Action_dim | 2 | | Action_dim | 2 |
| State_dim | 48 | | State_dim | 36 |
| Gamma | 0.99 | | Gamma | 0.99 |
| **DDPG** | | | **DDQN** | |
| Learning rate | 1e-3 | | Learning rate | 1e-4 |
| Policy Network | 256,256 | | Policy Network | 64,64 |
| Replay memory size | 20000 | | Update steps | 500000 |
| Update steps | 20000 | | | |
| **VOCE** | | | **CQL** | |
| Learning rate | 1e-3 | | Learning rate | 1e-4 |
| Policy Network | 256,256 | | Policy Network | 64,64 |
| Alpha scale | 10 | | Update steps | 500000 |
| KL constraint | 0.01 | | Alphas | 0.05,0.1,0.5,1,2 |
| Dual constraint | 0.1 | | | |
| Update steps | 4000 | | | |
| **CopiDICE** | | | | |
| Learning rate | 1e-4 | | | |
| Policy Network | 256,256 | | | |
| Alpha | 0.5 | | | |
| Cost limit | 10 | | | |
| Update steps | 100000 | | | |
| **BCQ-Lag** | | | | |
| Learning rate | 1e-3 | | | |
| Policy Network | 256,256 | | | |
| Cost limit | 10 | | | |
| Lambda | 0.75 | | | |
| Beta | 0.5 | | | |
| Update steps | 100000 | | | |

