# OpenReview forum: "Offline Inverse Constrained Reinforcement Learning for Safe-Critical Decision Making in Healthcare"
_ICLR.cc/2025/Conference — Submitted to ICLR 2025_

### Official Review · Reviewer_7wPo · 2024-10-31

**Soundness:** 3
**Presentation:** 2
**Contribution:** 2
**Rating:** 5
**Confidence:** 4

**Summary:**

This paper proposes an Offline Inverse Constrained RL method (ICRL) for medical applications, which aims to (1) learn decision-making constraints from observational data using a history-dependent model architecture (‘Constraint Transformer’, CT) as well as simulated unsafe decision-making trajectories, and (2) using these within a constrained offline RL optimization procedure.
The goal is to avoid unsafe behavior in the output policy.

**Strengths:**

- The paper addresses the interesting problem of learning safe decision-making policies from observational data, which is an important challenge for applying reinforcement learning to healthcare applications.
- Considering the inherent challenge of evaluating such methods, authors consider multiple metrics to assess the performance of policies trained with their method / different baselines.

**Weaknesses:**

1. The method proposed (and in particular its complexity, as it involves a range of different modeling steps) needs to be better motivated. Below are some example modeling choices that would require a more in-depth explanation + detailed experimental analysis:
- Is the generative model in L302 not biased by partial observability? As the policy changes from expert to violating policy, does this not induce confounding in the generative model?
- The fact that medical decision-making problems are history-dependent or POMDPs is not new (Yu et al., 2020, Reinforcement Learning in Healthcare: A Survey), yet it is presented as a major contribution.
- Authors find that the cost function correlates with the mortality rate. Isn’t this expected from the fact that doctors’ policy aims to avoid mortality? Is the cost function truly learning something distinct from the reward function, or is it just providing additional reward shaping based on imitation learning? Table 4 would benefit from additional results comparing to TD3+BC (RL and imitation learning regularization) and to CDT without cost or with custom cost.

2. Experiment results do not demonstrate a clear added value for the proposed method, considering its added complexity.
- Evaluation: constraints are inferred and optimized, but we have no ground-truth knowledge of what they should look like. I believe this could be evaluated in a toy experiment.
- Overlapping results: (1) Table 2: CDT with and without CT cost overlap. What is the added value of CT, considering the significant additional modeling complexity required? (2) Fig 7: Authors claim a difference at DIFF=0 between their methods and the baseline, but both curves overlap in confidence interval.
- How does CDT without cost work -- is it simply a decision transformer? If so, why does it perform so much better than DDPG?
- L504 Why is CDT+CT producing a policy that is more “distant from the clinician’s policy” than other offline RL baselines? Are we not specifically learning constraints based on what actions are *different* from the behavioral policy -- ie doing a form of regularization to imitation learning?


3. Clarity is poor overall, and this observation relates to my multiple questions above. Some additional comments:
- Is $\zeta_{\theta}$ dependent on $\tau$ or (s,a)?
- Vertical white space is reduced excessively throughout the paper, which makes for an unpleasant reading experience.
- L321 where does a_t come from, is it sampled from the dataset D_e? So is the objective simply imitation learning? Or is there a missing reward signal to make it an RL objective?
- what is the strategy for bolding results in Table 2, when they clearly overlap with others? Same comment for red numbers in Table 4.
- Table 2: How does the ‘generative model’ output a policy?
- Table 3: how can proportions for CDT + CT be 0% if at least one value exceeds the thresholds considered? Is 0% an approximation?
- Fig 7: How is mortality estimated here?

4. Additional details
- L262: Transformers are not ‘interpretable’ per se beyond the first layer.

**Questions:**

See above.
- How does the evaluation procedure described in L190 relate to off-policy evaluation methods / propensity-weighting schemes?
- Fig 1: what is a state?

---

> ### Author Response · Authors · 2024-11-24
>
> ## 1. Method
>
> * Is the generative model in L302 not biased by partial observability? As the policy changes from expert to violating policy, does this not induce confounding in the generative model?
>
>   We acknowledge that partial observability may introduce bias to the generative model. In our approach, the input to the generative model is state variables derived from expert data. However, these variables may not fully capture the true underlying state of the patient, potentially leading to biased estimations of the actual state. To mitigate the issue of partial observability, we incorporate historical sequence information. Moreover, the objective of the generative model is not to perfectly replicate the true state but to enhance data diversity by introducing specific "violating behaviors."
>
>
>   To verify the reliability of the patient trajectories generated by the model, we compared the generated data with the real data, including patient indicator accuracy and behavioral distribution analysis, as shown in Figures 11 and 12. The results indicate that the generated data align with statistical distributions and behavioral patterns consistent with medical theoretical knowledge.
>
> * The fact that medical decision-making problems are history-dependent or POMDPs is not new (Yu et al., 2020, Reinforcement Learning in Healthcare: A Survey), yet it is presented as a major contribution.
>
>   Thank you for your valuable feedback. We understand your point that the history dependency or partially observable Markov decision processes (POMDPs) in medical decision-making are not novel concepts. In our study, our primary contribution is not merely highlighting this issue but rather designing an inverse constrained reinforcement learning (ICRL) method specifically tailored for safe decision-making in medical scenarios. We are also the first to propose an ICRL method tailored for healthcare scenarios, aimed at ensuring safety in medical decision-making.
>
>   Specifically, our work focuses on how to use the ICRL approach to learn safety constraints in medical contexts and leverage these constraints to guide safety-first decision-making. This approach is designed to address the practical needs of the healthcare domain, rather than simply reiterating the importance of history dependency. We will make this distinction more explicit in our revised manuscript to better highlight the contributions of our work.
>
> * Is the cost function truly learning something distinct from the reward function, or is it just providing additional reward shaping based on imitation learning?
>
>   Thank you for the valuable feedback. We understand your concerns. To clarify, we used the same simple reward function in each experiment. For example, in the sepsis task, the reward function (as shown in equation 9) is only related to the SOFA score. However, our constraint function can learn more information beyond the SOFA score. For instance, the constraint function can learn metrics such as heart rate, lactate levels, and others unrelated to SOFA (as shown in Figure 5). It can also capture hidden state information, such as NEWS2 and MAP (as shown in Figure 14), which are not explicitly included in the state but can be effectively identified by the constraint function as signals of danger, helping guide decisions.
>
>   Therefore, the constraint function is not merely an additional reward shaping to the reward function but can learn and supplement key clinical risk-related information, thereby improving the safety and decision quality of the model.
>
>   Based on the reviewer’s suggestions, we have added comparative experiments to Table 4, including TD3+BC, CDT without cost, and CDT with custom cost. The experimental results demonstrate that our proposed CDT+CT framework still significantly outperforms other methods across multiple key metrics. Thank you for your valuable feedback.

---

> ### Author Response · Authors · 2024-11-24
>
> ## 2.Experiment results
>
> * **Toy experiment:** Thank you for your valuable feedback. We agree that evaluating the inference and optimization of constraints in a controlled toy environment is highly significant, as it can provide valuable insights into the interpretability and effectiveness of constraint learning. To the best of our knowledge, there are currently no publicly available toy environments suitable for healthcare scenarios.
>
>    Moreover, we have considered creating a toy environment specifically for medical scenarios, such as simulating key aspects of sepsis management. However, designing such an environment faces significant challenges, including defining reasonable state transition equations, modeling realistic constraints, and providing reliable offline expert datasets based on the environment. Expert datasets play a critical role in validating inverse constrained reinforcement learning (ICRL) methods. Due to these limitations, we have not yet conducted a suitable toy experiment.
>
>    We fully understand the value of your suggestion and plan to further explore potential toy environments in future research to complement real-world scenario evaluations and provide additional research insights.
>
> * Overlapping results:
>
>   **(1) What is the added value of CT, considering the significant additional modeling complexity required?** The significant superiority of CDT+CT. The results in Table 2 demonstrate that CDT+CT clearly outperforms CDT (without CT) across multiple metrics. This indicates that the inclusion of CT plays a critical role in optimizing the safety of the policy, rather than being merely an additional modeling complexity. In medical scenarios, we believe that the modeling cost of incorporating CT is far outweighed by the safety value it provides. Compared to the potential risks in clinical decision-making, CT offers stronger safety guarantees, effectively reducing decision-making risks and ensuring the applicability of the policy in complex medical environments.
>
>   **(2)Fig 7: Authors claim a difference at DIFF=0 between their methods and the baseline, but both curves overlap in confidence interval.** Figure 7 illustrates the correlation between the observed patient mortality rate and the difference between the dosage recommended by the learned optimal policy and the actual dosage administered by clinicians (i.e., the clinician policy). It demonstrates that when there is no difference between the optimal policy and the clinician policy, the lowest mortality rate is observed. This indicates that patients receiving dosages similar to those recommended by the optimal policy have the lowest mortality rate, suggesting that the learned policy is effective [1]. Both the DDPG and CDT+CT methods are effective.
>
>   Generally, when the difference is zero (DIFF=0), lower mortality rates are observed, indicating that when clinicians act according to the optimal policy, higher patient survival rates are achieved [2]. From this perspective, CDT+CT outperforms the DDPG method.
>
>
> * **why does it perform so much better than DDPG?**
>
>   CDT without cost works as a decision transformer with randomness. The Decision Transformer [3] performs better than DDPG, which indirectly supports the advantage of using historical information in medical decision-making.
>
>
> * **Why is CDT+CT producing a policy that is more “distant from the clinician’s policy” than other offline RL baselines?**
>
>   Clarification on “distance from the clinician’s policy”: We acknowledge that there is some ambiguity in the original text regarding this point. While safer and more conservative policies align with clinical guidelines, they do not necessarily reduce the statistical difference from clinician behavior (such as RMSE or F1 score). The higher RMSE reflects the design philosophy of the CDT+CT strategy, which prioritizes safety over fully mimicking clinician behavior, especially when clinician decisions exceed the learned safety constraints. What needs to be reinterpreted is that "safer and more conservative policies" are intended to optimize safety and constraint compliance, rather than minimizing the statistical difference from clinician behavior (e.g., RMSE or F1).
>
> [1] Raghu A, Komorowski M, Celi L A, et al. Continuous state-space models for optimal sepsis treatment: a deep reinforcement learning approach[C]//Machine Learning for Healthcare Conference. PMLR, 2017: 147-163.
>
> [2] Jia Y, Burden J, Lawton T, et al. Safe reinforcement learning for sepsis treatment[C]//2020 IEEE International conference on healthcare informatics (ICHI). IEEE, 2020: 1-7.
>
> [3] Chen L, Lu K, Rajeswaran A, et al. Decision transformer: Reinforcement learning via sequence modeling[J]. Advances in neural information processing systems, 2021, 34: 15084-15097.

---

> ### Author Response · Authors · 2024-11-24
>
> ## 3. Presentation
>
> * **Is ζθ dependent on τ or (s,a)?**
>
>   ζθ dependents on τ, where τ={s0, a0, s1, ...}. If the length of τ is 1, τ = {s0,a0}.
>
> * **Vertical white space is reduced excessively throughout the paper, which makes for an unpleasant reading experience.**
>
>    Thank you for your feedback. We understand that excessively reducing vertical white space can negatively impact the reading experience. In the revised version, we will appropriately increase the spacing between paragraphs to enhance the document's readability and visual appeal.
>
> * **L321 where does a_t come from, is it sampled from the dataset D_e? So is the objective simply imitation learning? Or is there a missing reward signal to make it an RL objective?**
>
>   Thank you for your question. To clarify, a_t is indeed sampled from the dataset D_e, and the reward signal \hat{R}_t in o_t (Line 311) is used to drive the RL objective, rather than being a simple imitation learning approach.
>
> * **what is the strategy for bolding results in Table 2, when they clearly overlap with others? Same comment for red numbers in Table 4.**
>
>    Thank you for your feedback, and we apologize for any reading inconvenience caused. In Table 2, the bolded numbers indicate which constraint performs better within the same model group when different constraints are applied. In Table 4, the red numbers highlight which method in the corresponding row performs better on the given metric.
>
> * **Table 2: How does the ‘generative model’ output a policy?**
>
>    In L315, we introduced the "policy model" in the generative model, which is used to simulate the generation of the "violating policy." We generate the policy using this model and conduct tests, ultimately obtaining the results presented in Table 2.
>
> * **Table 3: how can proportions for CDT + CT be 0% if at least one value exceeds the thresholds considered? Is 0% an approximation?**
>
>    In Table 3, the CDT+CT method does not generate drug doses that exceed the considered thresholds. Therefore, the 0% proportion indicates that no doses exceeding the thresholds were generated by this method, rather than being an approximation. This demonstrates the effectiveness of the method in ensuring safety by successfully avoiding drug doses that exceed the established thresholds.
>
> * **Fig 7: How is mortality estimated here?**
>
>    In the real medical dataset, we know the patient's state and the drug dosage (a) under the doctor's policy. We use the estimated policy to provide the drug dosage (b) for the same patient state under the estimated policy. We then calculate the DIFF for each patient state, which is b−a, and we also compute the mortality rate and standard deviation (std) of patients under the doctor's policy for different DIFF values, thereby obtaining Figure 7.
>
> ## 4. Additional details
>
> * **L262: Transformers are not ‘interpretable’ per se beyond the first layer.**
>
>   Thank you for the reviewer’s feedback. We understand your point, and indeed, the interpretability of Transformer models may decrease at deeper layers. However, when we mentioned "better model interpretability," we were specifically referring to the Transformer model in comparison to traditional LSTMs. In the revised manuscript, we will further clarify this point, emphasizing that the interpretability of Transformer models is primarily evident in the first layer and the overall structure, rather than in the deep internal representations.

---

> > ### Comment · Reviewer_7wPo · 2024-11-25
> >
> > Dear Authors,
> >
> > Thanks for your rebuttal. For future reference, I recommend highlighting changes to the manuscript in a different color to help reviewers know where to look. The presentation of the paper is still much too crowded and confusing to parse. Table 4 is now so small it’s illegible.
> >
> > I still recommend rejection. Many of the claims made in the paper are simply not supported by experimental evidence nor motivated by a rigorous framework. On the empirical side, most importantly, CDT + no cost performs on par with CDT+CT. Most results are overlapping, despite authors still suggesting their method as superior with red/blue/bold highlights. This defeats the purpose of the proposed method, considering its complexity. I also find the performance of the TD3+BC baseline very poor considering it usually outperforms CQL on offline RL tasks.
> >
> > Regarding a toy experiment, the point is not to use another paper’s simulation (even though these exist, see for example: Oberst and Sontag, 2019, Counterfactual off-policy evaluation with gumbel-max structural causal models), but to design a simple setup in which you can verify that your algorithm works as expected considering the assumptions you have made. This would not be challenging if a clear problem formalism had been established, but echoing Reviewer 4ZYT's comments, this is not proposed in the paper.
> >
> > As for the lack of rigorous motivation, from the formulation of the cost function learning problem, it looks like you are implementing a form of imitation learning regularization. This is a well-studied approach in offline reinforcement with real theoretical backing (e.g. Jin et al., 2021, Is Pessimism Provably Efficient for Offline RL?) and extensions to the partially-observable setup. I still struggle to understand in what way the medical setting is different and requires this alternative method.

---

### Official Review · Reviewer_2zeG · 2024-11-03

**Soundness:** 3
**Presentation:** 3
**Contribution:** 3
**Rating:** 6
**Confidence:** 5

**Summary:**

This paper introduces the Constraint Transformer (CT) framework to enhance safe decision-making in healthcare. The proposed CT model uses transformers to incorporate historical patient data into constraint modelling and employs a generative world model to create exploratory data for offline RL training. The authors supported their points by presenting experimental results in scenarios like sepsis treatment, showing that CT effectively reduces unsafe behaviours and approximates lower mortality rates, outperforming existing methods in both safety and interoperability.

**Strengths:**

This paper shows its strengths in the following aspects:

The paper addresses the novel angle of ensuring safety in offline reinforcement learning (RL) for healthcare, a critical and previously underexplored issue.

It incorporates Inverse Constrained Reinforcement Learning (ICRL) into offline reinforcement learning (RL) for healthcare, introducing a novel approach to inferring constraints from expert demonstrations in a non-interactive environment.

The implementation of a causal transformer to learn the constraint function is interesting, allowing the integration of historical patient data and capturing critical states more effectively.

Extensive results on 2 well-known dynamic treatment regime datasets are presented. The proposed Constraint Transformer (CT) framework is shown to reduce unsafe behaviours and approximates lower mortality rates.

**Weaknesses:**

I think the paper is generally solid and I vote for its acceptance. However, there are quite a lot of presentation issues and technical details to be addressed.

1. result

The RMSE_VASO of the proposed method is greatly larger than that of other policies. Also, it seems that CDT+CT outperforms other baselines on the importance sampling-based method but not on DR. Could you explain why? The authors said, "Since CDT+CT produces safer and more conservative policies, there is a certain distance from the clinician’s policy, so it does not perform as well in terms of RMSEvaso and F1 score." From my understanding, more safe and conservative policies should result in a smaller distance from the clinician's policy. Is this statement inconsistent with the result?

2. presentation issue

This paper has multiple presentation issues. Due to the space limit, I will only point out a few of them. *a)* Figure 1 is quite confusing. If an agent learns stochastic policy, its action choice can still be distributional. The authors misunderstood some fundamental concepts of MDPs in RL. *b)* "The Markov decision is not compatible with medical decisions." is technically wrong. The Markov decision **IS** compatible with medical decisions if you define the MDP correctly. To do RL, we must use the MDP assumption in a certain way. If I understand the authors correctly, you meant to say 'strictly first-order Markov' is not a fit for medical decisions. But why bother mention it in the introduction? Wouldn't a RNN solve this problem? *c)* Line 63-77: these 2 challenges are not in ICRL, but in applying ICRL in healthcare.

**Questions:**

NA

---

> ### Author Response · Authors · 2024-11-24
>
> ## 1. Result.
>
> **Thank you for the reviewer’s comments.** We have further considered the metrics in Table 4: From the work [1], we learned that the DR evaluation method relies on an accurate estimation of the model-based value function. Due to the presence of behavior policy and value approximation errors, it may not be doubly robust, which can lead to a loss of reliability. As a result, we no longer use DR as one of the evaluation metrics.
>
> **Clarification on “distance from the clinician’s policy”:** We acknowledge that the RMSE_VASO of the proposed method is significantly larger than that of other policies. This outcome can be attributed to the fact that safer and more conservative policies prioritize optimizing safety and constraint compliance over minimizing the statistical difference from clinician behavior. As a result, while these policies may perform better in terms of safety and constraint adherence, they may not align closely with the clinician’s decisions, which can lead to a higher RMSE_VASO. As for the statement, "more safe and conservative policies should result in a smaller distance from the clinician's policy," we understand the potential confusion. However, while safer policies are indeed more conservative in their actions, they may deviate more from the clinician’s typical behavior due to the stricter adherence to safety constraints. In contrast, less conservative policies might better align with clinician decisions, even though they may not fully prioritize safety. Therefore, there is a trade-off between safety, constraint adherence, and the statistical distance from clinician behavior.
>
> We hope this clarifies the apparent inconsistency, and we will revise the manuscript to reflect this distinction more clearly. Thank you again for your valuable feedback.
>
> [1] Luo Z, Pan Y, Watkinson P, et al. Position: reinforcement learning in dynamic treatment regimes needs critical reexamination[J]. 2024.
>
> ## 2. presentation issue
>
> **a)  Figure 1:** We understand the confusion. Indeed, in reinforcement learning, an agent can learn a stochastic policy, where the action selection is based on a probability distribution over possible actions for a given state, rather than a single fixed action. The representation in Figure 1 might not have accurately conveyed this point. We would like to clarify that the purpose of Figure 1 is to show that in real clinical scenarios, a physician's decisions may involve some degree of randomness or uncertainty, influenced by various factors such as the patient's clinical history, personal experience, intuition, etc. However, in the RL model, the agent makes decisions based on the current patient state. If the agent follows a deterministic policy, it will select a fixed action for the given state; if it follows a stochastic policy, the agent's behavior will be chosen according to a probability distribution, but this does not mean that it considers the patient’s historical information to make a “suitable” random decision.  Considering that Figure 1 does not accurately convey the information, we have decided to remove it in the revised version.
>
> **b) strictly first-order Markov:** Thank you for your insightful feedback. We understand your point that the Markov decision process (MDP) can indeed be compatible with medical decisions if defined correctly. In our previous statement, we considered the general interpretation of Markov processes, which is typically assumed to be strictly first-order Markov, without specifying any further constraints.
>
> Moreover, the existing ICRL methods are based on the assumption of a "strictly first-order Markov" process, which we believe is not well-suited for medical decision-making. Given challenge 2 in medical scenarios, we argue that the current ICRL methods cannot be directly applied.
>
> Regarding your suggestion about RNNs as a potential solution, we have indeed discussed this in our methodology and conducted a comparative analysis in the experimental section (e.g., Table 9). We found that attention-based methods, such as Transformers, are more effective at learning historical information in medical scenarios. The reason why Transformer demonstrates greater advantages in medical scenarios is also a topic worth discussing in future research.
>
> c. We apologize for the potential misunderstanding in our original text. We should have clarified that the challenges mentioned are related to the application of ICRL in the healthcare domain, rather than being inherent challenges of ICRL.

---

> ### Comment · Reviewer_2zeG · 2024-11-24
> **Concern on the RMSE_VASO result**
>
> Thanks to the authors for their clarifications. Besides the Figures and MDP definition, there are still tremendous tiny presentation errors, such as 'more safe' rather than 'safer'. The paper is not impossible to read, but I hope the authors can put more effort into improving the readability and clarity.
>
> Besides, I'm still not quite convinced by the statement that 'the model weights safety over minimising the distance to a clinical policy'. If so, both the RMSE_VASO **and** RMSE_IV should be larger. In fact, we only found RMSE_VASO looks significantly larger. Therefore, it is reasonable to suspect that:
>
> a) the model is constrained in a biased way. Evidence can be found in Figure 5, the lactate subfigure, where the lactate curve is nearly flat. From a medical perspective, vasopressor is strongly correlated with lactate. lactate>2 is considered abnormal, and lactate>4 is considered very dangerous. However, this medical convention cannot be verified in the plot.
>
> b) The world model for evaluation is biased, as other reviewers also mentioned. The biased world model likely affects model selection during training and, therefore, favours the model 'punish less/too much about vasopressor'.
>
> I hope the authors carefully review these and clarify the reason behind them.
>
> Overall, I appreciate the methodology proposed. We (including other reviewers) should all admit that learning a medical model will certainly include bias and, in most cases, much more severe than in any other field. However, I still wish these problems to be discussed and make sure the credit is not overclaimed.

---

### Official Review · Reviewer_4ZYT · 2024-11-03

**Soundness:** 2
**Presentation:** 2
**Contribution:** 2
**Rating:** 3
**Confidence:** 5

**Summary:**

The paper presents an offline approach for inverse RL in constrained settings for safety critical decision making. The overall premise of the paper is the traditional RL methods might overlook common-sense constraints which might lead to potentially harmful decisions in a high-risk setting. Moreover, such healthcare decisions must usually be inferred from offline batch data, where there is no online/limited interaction with the environment and the exact constraints might be unknown as is typically the case in healthcare. The authors introduce a two step approach to overcome these issues: i) first they employ an attention-based architecture to identify long-range dependencies and critical aspects of a patient's history; ii) based on the learnt representation from i), the authors use a generative model of the environment for data augmentation that enables unrolling/simulating potentially unsafe decision sequences. The authors try demonstrate on MIMIC III that their approach recovers the relevant constraints accurately, is performant in offline settings and enables safe policy learning.

**Strengths:**

Overall I think the paper tries to address a very relevant and significant problem that is not only limited to healthcare. The paper is written well in general and I like the way the experiments are laid out in the sense that the authors clearly divide their analysis into questions that tackle various aspects of the evaluation. The authors also do a good job in motivating the problem overall.

**Weaknesses:**

1) The authors dont make explicit what their assumptions are and I had to do a lot of sifting through details multiple times, to get a clearer picture of the exact problem setting: it is assumed that there is a reward and a set of constraints (costs) within the MDP; the reward is observable but the constraints are unknown which is why an IRL approach is warranted.

2) The authors also assume that the expert is optimal which may not be the case in practice so the authors should make this explicit in an assumptions section. The authors should in particular make very explicit what the distribution of clinician actions is for their applications and allow us to visualise this. For instance, if the most common clinician action choice is to provide very small doses of vasopressors and fluids, how does this impact the quality of the constraints learnt? I would think that if the distribution of clinician actions is strongly biased towards certain decisions, this might have significant impact on the constraints learnt/might end up overconstraining the problem or end up not learning anything meaningful.

3) My main criticism for this paper is I feel the method contains a LOT of components and it is not always clear what the purpose of each component is. While the purpose of the transformer is to retain relevant aspects of the patient history and capture critical states, I would have liked to have seen some analysis into what is learnt/whether these points are in fact critical, why they are important etc. There is a huge body of work on learning relevant representations e.g. through decision/task-focused learning (Sharma et al 2024 https://arxiv.org/pdf/2110.13221) or otherwise through information-based methods (Liu et al 2024, https://arxiv.org/pdf/2405.09308) or similar, but the authors dont compare the representations they learn to any other method which makes it hard to understand whether what is captured is indeed relevant. There is also a large amount of work related to capturing decision-regions or important points in a patient trajectory that might impact doctor decision making in terms of how much they agree on what to do (Zhang et al 2022 https://www.nature.com/articles/s41746-022-00708-4) which should be discussed here. If nothing else, the authors should compare what is learnt to a standard LSTM representation. Subsequently, a generative model is used to unroll or simulate downstream decision sequences and perform safe decision-making under constraints. There is little discussion on how the simulator behaves as a causal model and serves as an adequate means of simulating possible trajectories. I will elaborate on this next.

4) The authors also mention that the generative model used to simulate trajectories is causal in the abstract and elsewhere. How can this be verified? What makes the representation causal? Is it robust against confounding biases? Are there experiments that validate this where you demonstrate explicitly that the effects of such biases are minimised and do not have downstream consequences on the constraints learnt and the simulations produced? Did you validate the simulations with domain expertise to check whether these were indeed valid potential trajectories/simulations of how a patient might evolve?

5) Throughout the paper, there is extensive mention of the terms Safe RL yet the entire literature on safe RL is ignored/not discussed meaningfully in terms of related work or in terms of establishing baselines for the approach to compare to (see Garcia et al 2015 https://www.jmlr.org/papers/volume16/garcia15a/garcia15a.pdf as a starting point). I understand the difference is now you are learning C but one could think of C as being part of a multiobjective reward R in which case all the works on multi-objective RL need to be discussed in a lot more detail in the related work. Given that there are various notions of what it means to be safe e.g. risk sensitive, risk-directed/aware etc. Where does this approach sit?

6) The authors dont provide any other experiments apart from MIMIC III. It would be good to have either some toy demonstrations to validate the utility of each component of the method and why it serves a purpose or to have another clinical data set with another clinical task where you demonstrate the performance of the approach (hopefully where the action distribution differs significantly)

**Questions:**

1) What are your assumptions in this work and why are they plausible?
2) How does the distribution of clinician actions impact/bias the work? If the clinician decisions are all centred around administering very few drugs does that not overconstrain the system or end up with you not learning any meaningful constraints at all? How do you deal with uneven clinician action distributions?
3) What is the purpose of each component in the story and what role does it play? Why dont you compare the representations you learn with other representation learning approaches.
4) What makes the learnt representation causal? Is it robust against various biases? Do you have clinical validation to support what you have learnt?
5) What role does the Safe RL literature play here. How do you compare to these?
6) Do you have another domain in which you can validate and support your findings?

---

> ### Author Response · Authors · 2024-11-24
>
> ## 1. What are your assumptions in this work and why are they plausible?
>
> Thank you for your question regarding our assumptions. We have summarized them as follows:
>
> The policy capable of treating patients to survive is the expert policy, but we do not require the expert strategy to be optimal. Our goal is to align as closely as possible with the expert policy to ensure safety and rationality during the learning process.
>
> The reward function is designed by experts and is observable. It is defined based on patient treatment outcomes (e.g., survival rates), which aligns with the common assumptions in reinforcement learning research in healthcare. For instance, many studies [1-3] use expert-designed reward functions as the foundation for optimization, and we adopt the same approach without modifying the definition of the reward.
>
> The constraint function, however, is unknown but implicitly influences physicians' decision-making. This assumption stems from real-world medical scenarios where doctors consider implicit constraints, such as patient risks and medication side effects, during decision-making. These constraints are often challenging to explicitly articulate or quantify. Therefore, using inverse constrained reinforcement learning to infer the constraints is both reasonable and necessary.
>
> [1] Raghu A, Komorowski M, Celi L A, et al. Continuous state-space models for optimal sepsis treatment: a deep reinforcement learning approach[C]//Machine Learning for Healthcare Conference. PMLR, 2017: 147-163.
>
> [2] Huang Y, Cao R, Rahmani A. Reinforcement learning for sepsis treatment: A continuous action space solution[C]//Machine Learning for Healthcare Conference. PMLR, 2022: 631-647.
>
> [3] Komorowski M, Celi L A, Badawi O, et al. The artificial intelligence clinician learns optimal treatment strategies for sepsis in intensive care[J]. Nature medicine, 2018, 24(11): 1716-1720.
>
> ## 2. How does the distribution of clinician actions impact/bias the work? If the clinician decisions are all centered around administering very few drugs does that not overconstrain the system or end up with you not learning any meaningful constraints at all? How do you deal with uneven clinician action distributions?
>
> Thank you for your valuable feedback. We understand your concern, and indeed, assuming that the expert strategy is optimal may not always hold true in practice. To address this, we have further clarified this point in the paper.
>
> Regarding the impact of clinician action distributions, we conducted tests on another clinical task (mechanical ventilation task), where the dataset contains a clinician action distribution different from that of the sepsis task (as shown in Figures 7 and 8). By comparing the experimental results in the mechanical ventilation task (see Table 8), we found that even with significant differences in clinician action distributions, the model was still able to achieve good learning results. This indicates that the clinician's action distribution does not significantly impact the learning process in our work. Therefore, we believe that even with an uneven action distribution, the model can still effectively learn the constraints.

---

> ### Author Response · Authors · 2024-11-24
>
> ## 3. 1) What is the purpose of each component in the story and what role does it play?
>
> Thank you for your feedback on the multiple components of our model. We agree that it is important to clarify the role of each component. Our model is designed to address specific challenges in medical decision-making through the following components:
>
> * Constraint Transformer: This component is used to learn constraint information in medical scenarios. By leveraging a causal attention mechanism, it captures relevant historical treatments and observations, ensuring the model focuses on critical parts of the patient trajectory. This addresses the non-Markovian nature of medical data.
>
> * Model-Based Offline RL: Since we only have expert datasets in real-world settings, we need data containing violations to train the Constraint Transformer. The Model-Based Offline RL serves as a generative world model that performs exploratory data augmentation. By simulating unsafe decision sequences, it enables the model to learn from these unsafe scenarios and avoid them in real applications.
>
> * Relationship between Components: The Model-Based Offline RL generates violating data, which is then used to train the Constraint Transformer.
> Responses to Specific Concerns
>
> **i. Analysis of “what is learnt/whether these points are in fact critical”:**
>
> We analyzed the relationship between patient multidimensional metrics and penalty values (as shown in Figure 6). The results indicate that the constraint function successfully captures "unsafe" states by assigning higher penalties. From this analysis, we can infer that the Constraint Transformer identifies critical states consistent with medical theory and expert knowledge.
>
> **ii. Comparison with standard LSTM representation:**
>
> Without violating data generated by the model-based approach, training the Constraint Transformer would not be feasible. As such, it is challenging to follow the suggestion of comparing the learned knowledge directly to a standard LSTM without this data. Instead, we compared the historical capture architecture (attention mechanism) of the Constraint Transformer with an LSTM by replacing the attention module with an LSTM. The results, presented in Table 9, demonstrate that the Constraint Transformer outperforms the LSTM-based approach.
>
> ## 3. 2) Why dont you compare the representations you learn with other representation learning approaches?
>
> Thank you for your valuable feedback. We fully acknowledge the importance of representation learning and recognize the significant success achieved by task-driven or information-based methods in various domains. However, our primary objective is to design a model that can learn domain-specific constraints in the medical field. Unlike general representation learning tasks, our goal is not just to learn task-relevant representations but also to ensure compliance with critical safety constraints in medical decision-making, such as drug dosage limits and fluid usage thresholds. These constraints are crucial to the learning process as they directly impact the safety and efficacy of treatment decisions.
>
> In the current framework, traditional representation learning methods may not effectively capture and learn these domain-specific constraints. To evaluate the effectiveness of our model, we compressed patient histories into a two-dimensional space and labeled patient states as “safe” or “unsafe.” We observed that the original patient metrics struggled to distinguish between safe and unsafe regions. In contrast, the embeddings produced by the CT layer, when mapped to two dimensions, successfully separated some of the safe and unsafe regions, as shown in Figure. 18.
>
> These results demonstrate that our method can learn meaningful representations related to medical constraints. We believe this analysis addresses your core concerns and further highlights the importance of incorporating constraint-aware representation learning in medical decision-making.

---

> ### Author Response · Authors · 2024-11-24
>
> ## 4. What makes the learnt representation causal? Is it robust against various biases? Do you have clinical validation to support what you have learnt?
>
> Thank you for the valuable comments on our work. Below is a detailed response to the reviewer's concerns:
>
> **What makes the learnt representation causal?**
>
> The causal nature of the generative model we mentioned in the paper is ensured by the use of a causal Transformer architecture. Specifically, during the calculation of each time step's representation, the model can only access the current time step and the previous inputs, without any access to future inputs. This causal structure guarantees the causality in the generative process. In practice, we implement causal masking to prevent any leakage of future information, thus ensuring the model adheres to the principle of causality. The specific reason why the Transformer has causality is not the focus of our research.
>
> **Is it robust against various biases?**
>
> To verify the robustness of the causal model, we conducted a sensitivity analysis. Specifically, we tested the impact of the target reward parameter on downstream outcomes, as this parameter significantly influences whether trajectories "violate" constraints. The results of the experiment show that an increase in the target reward indeed affects downstream results, but there is a threshold beyond which the impact of changes on the results stabilizes. The sensitivity analysis results are detailed in Appendix C.4 of the paper.
>
> **Do you have clinical validation to support what you have learnt?**
>
> For validating the clinical validity of the generated trajectories, we analyzed the distribution and reasonableness of the generated data. We confirmed that the generated trajectories not only "violate" the constraints in terms of the reward function (i.e., they incur a high penalty), but also that the clinical indicators for these trajectories remain within reasonable ranges. The experimental results can be found in Appendix C.2 and C.3 of the paper. In these experiments, we demonstrated that the generated trajectories align with expected clinical logic, ensuring that the simulated trajectories reflect plausible potential patient evolutions.
>
> ## 5. What role does the Safe RL literature play here. How do you compare to these?
>
> **Relationship with existing work in Safe Reinforcement Learning.** The approach we propose is not intended to directly compete with or replace existing safe reinforcement learning methods and multi-objective RL, but rather to serve as a complementary tool for safe RL. Specifically, while current safe RL methods focus on ensuring that the agent avoids dangerous behaviors during the learning process, our method focuses on learning implicit constraints that are difficult to express explicitly in medical settings. These constraints, such as patient risks, drug side effects, and other factors, are often difficult to quantify or directly represent in medical decision-making. Therefore, our goal is to learn these constraints to provide more reliable safety guarantees for safe RL.
>
> **Relationship with existing work in Multi-objective RL.** While our learned constraints (C) share some similarities with the decomposition of objectives in Multi-objective RL, our method does not directly optimize a multi-objective function. Instead, it focuses on extracting latent constraints from historical data to improve the safety of subsequent reinforcement learning strategies. The key distinction from traditional Multi-objective RL is that we treat the learned constraints as an auxiliary mechanism rather than a direct optimization objective.
>
> **Comparison with existing methods.** Since our method focuses on constraint learning rather than traditional safe RL methods for risk management or multi-objective optimization, our comparison targets existing constraint learning approaches, rather than safe RL methods. In the evaluation section of the paper, we treat constraint representation as a component and test it within multiple safe RL methods such as VOCE, CopiDICE, BCQ-Lag, and CDT.
>
> **Concept of safety in this work.** As pointed out by the reviewer, there are multiple notions of safety in the safe RL literature, such as risk-sensitive and risk-directed/aware. Our work focuses on ensuring the safety of policies in the medical domain through constraint learning, rather than directly modeling risk-sensitive or risk-directed/aware approaches.

---

> ### Author Response · Authors · 2024-11-24
>
> ## 6. Do you have another domain in which you can validate and support your findings?
>
> Thank you for the valuable comments regarding the experiments. In addition to the experiments conducted on the MIMIC-III sepsis task, we have also performed testing on a mechanical ventilation task. In this task, we used a discrete action space and visualized the action distribution, demonstrating the distribution of the data. We observed that the action distribution in the mechanical ventilation task differs significantly from that in the sepsis task, further validating the applicability of our method across different clinical tasks. Additionally, in the off-policy evaluation section, we referred to Table 8 in the original paper to further validate the effectiveness and practical utility of our method.
>
> Regarding toy demonstrations, we have not yet found suitable toy demonstrations for the healthcare domain, especially those that can effectively validate the role of each component in reinforcement learning. Moreover, we have considered creating a toy environment specifically for medical scenarios. However, designing such an environment faces significant challenges, including defining reasonable state transition equations, modeling realistic constraints, and providing reliable offline expert datasets based on the environment. Expert datasets play a critical role in validating ICRL. Due to these limitations, we have not yet conducted a suitable toy experiment. We fully understand the value of your suggestion and plan to further explore potential toy environments in future research to complement real-world scenario evaluations and provide additional research insights.

---

> > ### Comment · Reviewer_4ZYT · 2024-11-27
> > **Reviewer Response to Author Comments**
> >
> > I thank the authors for additional experiments shown in Figure 7 and 8 which have helped clarify my concerns re clinician bias. However, my concerns regarding lack of other domains or suitable toy demonstrations to showcase the key characteristics of the proposed solution have not been adequately addressed. I also think the authors should compare to constraint-based approaches and multiobjective RL even if the comparison is not a direct one. For instance, one might think of how the learned constraints might capture different information to MORL and show what this additional information might give us in terms of performance advantages or safety. One might also want to interpret what this additional information as as oppose to pre-specified objectives. I would also like to understand if the optimization of these learnt constraints is easier than optimizing for some given constraints. I am also not fully convinced by what makes the model 'causal'. I acknowledge the use of prior literature on causal transformers but I question the role confounding might play in the representation learnt. For these reasons, I maintain my score.

---

### Official Review · Reviewer_sS3W · 2024-11-04

**Soundness:** 3
**Presentation:** 3
**Contribution:** 3
**Rating:** 3
**Confidence:** 3

**Summary:**

This paper studies inverse RL with learnable constraints in the offline setting, focusing on practical applications in healthcare tasks. The main approach appears to be combining the decision transformer architecture in the offline RL literature with inverse constrained RL with max entropy framework. Experiments were conducted on two healthcare tasks: sepsis and mechanical ventilation.

**Strengths:**

This paper studies an interesting and potentially important problem.

**Weaknesses:**

There are some inconsistencies throughout the text in terms of mathematical descriptions, and the experimental evaluation also suffers from some issues.

**Questions:**

- Table 1: the definition of "too high" is somewhat problematic. Claiming "too high" is associated with increased mortality making it an "unsafe behavior" without conditioning on the patient state seems inappropriate. Even though the footnote says this should be patient-dependent, the table still shows a single threshold.
- Sec 3 problem formulation defines a non-Markov decision process (non-MDP) but the remaining text all references an MDP (L173-L175).
- L173: What does "the MDP derived by augmenting the original MDP with the network" mean exactly? Why do we augment the MDP with a network? Or do you mean augmented with the constraint?
- L176-L206 evaluation metric defined using "DIFF": this graphical analysis known as the "U-curve" is deprecated and widely criticized because it cannot distinguish a good policy from a bad policy (even a random one). See Figure 4 in Gottesman et al. "Evaluating Reinforcement Learning Algorithms in Observational Health Settings" (https://arxiv.org/abs/1805.12298)
  - also "We randomly select 2N patients from the dataset where N patients survived under the doctor’s treatment, and N patients died" do not seem justified. Depending on N this may not be representative of the dataset. Why not use some form of weighted average in the ranking?
- L244: what does "creating an offline RL for violating data" mean?
- L190: "probability of approaching the optimal policy" L438 "use ω to indicate the probability that the policy is optimal" - in offline setting, we can never obtain the optimal policy, so I don't believe this terminology is appropriate.

---

> ### Author Response · Authors · 2024-11-24
>
> ### 1. The definition of "too high".
>
> Thank you for your insightful comment regarding the definition of "too high" in Table 1. We agree that a single threshold might oversimplify the complexity of patient-dependent conditions, particularly when evaluating safety-related behaviors in clinical decision-making.
>
> To address this concern, we conducted additional experiments by stratifying patients based on their SOFA scores into three categories: mild, moderate, and severe sepsis, shown in Figure.9. Our findings reveal that the DDPG model tends to overestimate medication dosages for patients with mild and moderate sepsis. These patients, in many cases, do not require such high dosages, which aligns with your concern that defining "too high" without considering patient state may not accurately reflect unsafe behaviors.
> We have updated Table 1 and the corresponding analysis to incorporate this patient-dependent stratification. This adjustment provides a more nuanced evaluation of unsafe behaviors and highlights the importance of adapting decision-making strategies to the specific clinical context of individual patients.
>
> We sincerely appreciate your suggestion, as it has helped us refine our analysis and improve the robustness of our study.
>
> ### 2. The definition of "non-MDP".
>
> We appreciate the reviewer’s comments regarding the Non-MDP framework defined in Section 3 and the MDP-based discussion in L173-L175. To ensure a more rigorous presentation, we have reformulated Constrained Non-MDP as a Constrained Partially Observable Markov Decision Process (Constrained POMDP) framework in Section 3 Problem Formulation. The reason we switch to CPOMDP is that it explicitly or implicitly models the underlying unobserved states (e.g., using latent variables to represent medical history), transforming certain seemingly "non-Markovian" characteristics into "partially observable" problems.
>
> However, existing Inverse Constrained Reinforcement Learning (ICRL) methods are only applicable to MDP scenarios. Therefore, in Lines 173–175, we discuss the MDP context of existing ICRL methods. The goal of this study is to attempt to extend the intuitive reasoning results of MDP-based ICRL methods to CPOMDP problems in medical scenarios.
>
> ### 3. Augmenting the MDP.
>
> Augmenting the MDP means introducing additional components that modify the original dynamics. This can include adding constraints, new variables, or in this case, a network (often a neural network). The idea behind this is to enrich or alter the agent's behavior in the environment, which could involve things like adding safety constraints, optimizing for efficiency, or introducing complex decision-making mechanisms.

---

> ### Author Response · Authors · 2024-11-24
>
> ### 4. Evaluation metric.
>
> i. Based on the reviewer's comments, we understand the criticism regarding the U-curve and have further analyzed the limitations of this method and our results. We have used random strategies and no-treatment strategies to plot the U-curve, and we observed that only the no-treatment strategy displayed a typical U-curve, while the random strategy did not show a clear U-curve. This observation indicates that not all random strategies result in a U-curve. Therefore, the U-curve can, to some extent, reflect the effectiveness of a strategy.
> Additionally, we believe that the point where DIFF = 0 in the U-curve is crucial, as it represents the situation where the strategy is exactly the same as the clinician's. At this point, if our strategy shows a lower mortality rate, it suggests that our strategy can identify superior strategies (e.g., strategies with lower mortality) from the clinician's approach. Therefore, we focus on this point and highlight that our strategy shows lower mortality at DIFF = 0, demonstrating that our method is capable of identifying better treatment strategies. This is the key insight we aim to present in the graph.
>
> In conclusion, our results do not solely rely on the U-curve to evaluate strategy performance. Instead, we use the mortality difference at DIFF = 0 to demonstrate the effectiveness of our strategy in better simulating and optimizing the clinician's approach. This analysis shows the potential of our strategy in practical applications, enhancing treatment outcomes and reducing patient mortality.
> We would like to thank the reviewer again for raising this question, which has allowed us to provide a more comprehensive evaluation and explanation of Figure 4.
>
> ii. Regarding the selection of patients in a 1:1 ratio of survivors to non-survivors, our rationale is based on the need to equally evaluate the treatment policies' impact on both survival and mortality cases. By doing so, we avoid the influence of imbalanced patient distributions in the dataset (e.g., where survived patients may be in the majority), which allows for a clearer assessment of the model’s ability to distinguish between effective and ineffective decisions. While using a weighted average based on the original distribution of the dataset is another approach, we believe this could lead to overestimating the performance on the majority class (such as survived patients), thereby obscuring the model’s performance on the minority class (such as deceased patients). This is crucial for identifying the model’s limitations and potential risks.
>
> ### 5. L244: what does "creating an offline RL for violating data" mean?
>
> To clarify, when we refer to "creating an offline RL for violating data," we are referring to the process of using a model-based offline reinforcement learning (RL) approach to generate data that represents situations where the treatment violates predefined safety thresholds (e.g., excessive drug dosages or inappropriate interventions). The purpose of this approach is to simulate and generate scenarios that reflect potential unsafe or suboptimal treatment decisions.
>
> ### 6. Revision about "the optimal policy".
>
> Thank you for your valuable feedback regarding the use of the term "optimal policy." We understand that in the offline setting, we cannot achieve the true optimal policy, and therefore, we agree that the terminology should more accurately reflect this reality.
> To address this concern, we propose the following revisions:
>
> L190: We revise the phrase "probability of approaching the optimal policy" to " aligning rate with the expert policy ".
>
> L438: We revise the phrase "use ω to indicate the probability that the policy is optimal" to "use ω to indicate the aligning rate with the expert policy ".
>
> Aligning rate is used to describe the degree of alignment between our policy and the expert policy (the policy capable of curing patients).

---

> > ### Comment · Reviewer_sS3W · 2024-11-26
> >
> > Hello, I appreciate the detailed response. As some of the other reviewers have also pointed out, there remains to be presentation issues even in the updated version. For example I found this sentence in L1152 "Corresponding experiments are conducted on the mechanical ventilator". Based on the rest of the paper I don't think the authors could realistically conduct experiments on an actual mechanical ventilator, but the sentence at face value claims so, and this is misleading.
> >
> > My main technical concern remains to be the evaluation metrics. It appears that the authors have invented several new metrics such as the DIFF plot (which I believe is the "U-curve analysis"), "alignment rate", which all seem to lack statistical rigor. In Fig 6, except for the random policy, all other three policies reach minimum mortality rate at DIFF of 0. For VASO DIFF, the green curve is almost identical to the black curve (no action) with the nearly identical minimum value of 0.14. Therefore, I still disagree that this is a valid evaluation method.
> > "1:1 ratio of survivors to non-survivors" could be justified as opposed to weighted averaging but the dataset construction should be need to be repeated multiple times to remove the effect of randomness.

---

### Official Review · Reviewer_rjzi · 2024-11-05

**Soundness:** 3
**Presentation:** 3
**Contribution:** 2
**Rating:** 6
**Confidence:** 2

**Summary:**

The paper proposed a constraint transformer method to tackle the problem that only historical data is provided in offline inverse reinforcement learning. The proposed method exhibits effectiveness in a sepsis task.

**Strengths:**

- The work is well motivated and investigates an important problem, where online test or interaction is unavailable and risky for ICRL.
- The paper is well written
- The proposed method achieved good performance on MIMIC-III derived dataset which is more realistic
- Limitation, especially that expert demonstrations are assumed as optimal, is well discussed.

**Weaknesses:**

- I’m curious any justification for only investigating one dataset for a specific task, sepsis treatment, though I agree sepsis is an important task and MIMIC-III has been broadly investigated. But given the proposed method is proposed targeting a general problem in healthcare, either more strictly scope the claims and investigated settings, or providing broader investigation could be helpful for further demonstrating the effectiveness of the proposed method.

**Questions:**

Please see weaknesses.

---

> ### Author Response · Authors · 2024-11-24
>
> Thank you for your valuable suggestions! Indeed, the current study focuses on the MIMIC-III dataset, primarily because sepsis and mechanical ventilation are well-established tasks within this dataset, which allows us to effectively demonstrate the performance and feasibility of the proposed method. We are aware that the method is intended to address a broader healthcare problem, and as such, the experiments conducted on MIMIC-III cover not only sepsis treatment but also the important task of mechanical ventilation (the definition in Appendix A.2 and the results in Figure 17, Table 8 and Table 9) to showcase the method’s adaptability across multiple tasks.
> Additionally, we will try to extend our research to other datasets, such as the eICU dataset [1], to further validate the method’s effectiveness and generalizability across different clinical settings.
>
> Thank you again for your suggestion—it will help us expand the scope and depth of our research!
>
> [1] Pollard TJ, Johnson AEW, Raffa JD, Celi LA, Mark RG and Badawi O. The eICU Collaborative Research Database, a freely available multi-center database for critical care research. Scientific Data (2018). DOI: http://dx.doi.org/10.1038/sdata.2018.178.

---

### Meta-Review · Area_Chair_jGFs · 2024-12-23

**Metareview:**

This paper proposes a transformer based model to overcome some challenges with safety in healthcare RL by using the decision-transformer framework combined with a reward design (IRL) setup where markovian assumptions may be violated. Empirical evaluation compares results in  sepsis (mechanical vent) management in MIMIC.

Reviewers have raised several concerns regarding, lack of clarity in claims and overall writing, insufficient empirical evaluation with experiments only demonstrating MIMIC-III, lack of comparison with other safe RL frameworks, including the claim of benefits of transformer based representation learning but lacking rigorous evaluation of other representation learning settings in healthcare RL, as well as choice and comprehensiveness of the evaluation measures.

Authors have clarified several concerns while also updating the draft. However, reviewers have lingering concerns that are not sufficiently addressed in the updated draft or the author response, therefore most reviewers have not updated their score. I therefore recommend a reject in broad consensus with reviewer concerns that there are still some claims that need to be well justified through writing, and rigorous empirical evaluation.

**Additional Comments On Reviewer Discussion:**

No additional points were raised during reviewer discussion.

---

### Decision · Program_Chairs · 2025-01-22

Reject